# Rationality and cognitive bias in captive gorillas' and orang-utans' economic decision-making

**Penelope Lacombe**[1]*, **Sarah Brocard**[1], **Klaus Zuberbühler**[1,2‡], **Christoph D. Dahl**[1,3,4‡]*

**1** Institute of Biology, University of Neuchatel, Neuchatel, Switzerland, **2** School of Psychology and Neuroscience, University of St Andrews, St Andrews, Scotland (United Kingdom), **3** Graduate Institute of Mind, Brain and Consciousness, Taipei Medical University, Taipei, Taiwan, **4** Brain and Consciousness Research Center, Shuang-Ho Hospital, Taipei Medical University, New Taipei City, Taiwan

‡ KZ and CDD are joint last authors.
* penelopelacombe@yahoo.fr (PL); christoph.d.dahl@gmail.com (CDD)

**Data Availability Statement:** All data are available (http://doi.org/10.5281/zenodo.4709798).

**Funding:** This work was supported with funding by the Swiss National Science Foundation (grant PZ00P3_154741 (CDD), 310030_185324 (KZ),

## Abstract

Human economic decision-making sometimes appears to be irrational. Partly, this is due to cognitive biases that can lead to suboptimal economic choices and context-dependent risk-preferences. A pertinent question is whether such biases are part of our evolutionary heritage or whether they are culturally acquired. To address this, we tested gorillas (*Gorilla gorilla gorilla*) and orang-utans (*Pongo abelii*) with two risk-assessment experiments that differed in how risk was presented. For both experiments, we found that subjects increased their preferences for the risky options as their expected gains increased, showing basic understanding of reward contingencies and rational decision-making. However, we also found consistent differences in risk proneness between the two experiments, as subjects were risk-neutral in one experiment and risk-prone in the other. We concluded that gorillas and orang-utans are economically rational but that their decisions can interact with pre-existing cognitive biases which modulates their risk-preference in context-dependent ways, explaining the variability of their risk-preference in previous literature.

## Introduction

Economic theories and mathematical modelling of decision-making are increasingly used to understand and predict human behaviour. Classic models, such as 'Expected Utility Theory' (EUT) [1], assume that humans are rational decision-makers, such that, when faced with a choice of options, they compare the utilities of all available options in order to gain the maximal utility [2]. 'Expected Utility Theory' is based on the premise that the value of an option is defined by its utility and that humans are capable of evaluating utilities to make well-deliberated decisions.

However, consistent experimental findings show that human economic behaviours often appear irrational, i.e., not aimed at maximising utility [3]. Investigating the factors that lead to seemingly or real irrational choices shows that the way utility is assessed can highly modulate

and NCCR Evolving Language (Agreement #51NF40_180888 (KZ)), and the Taipei Medical University (Startup-funding, grant 108-6402-004-112 (CDD)). The funders had no role in study design, data collection and analysis, decision to publish, or preparation of the manuscript.

**Competing interests:** The authors have declared that no competing interests exist.

decision-making. Indeed, assessing and comparing utilities appears to be a difficult process, and decision-makers rarely have a perfect knowledge of all options they have to choose from, which can lead to *apparently* irrational behaviours (i.e., subjects do not maximise utility). For instance, humans can forego high-profit options in order to explore unfamiliar alternative options, even if they risk ending up with less profit ([4, 5]). This explorative behaviour aims to evaluate the utilities of all options, which is necessary for the decision-maker, and is especially important in unstable environments where continued sampling of options is essential for success [6].

Thus, apparently irrational behaviours can sometimes be explained by sampling strategies (or sampling difficulties) aimed at evaluating utilities, and do not conflict with EUT.

However, a large part of the apparently irrational behaviours described in the literature show a more severe discrepancy between human decision-making and what is predicted by EUT (e.g., violation of the independence axiom of EUT [7]), which cannot be explained by issues in utility evaluation or perception. This led to the development of behavioural economics, and the establishment of 'Prospect Theory' [8] that has been particularly influential in explaining intuitive and emotional choices rather than rational aspirations to maximise gain.

'Prospect Theory' states that options are not only assessed by their (true) utilities but also compared to a reference as gains or losses, and losses are valued more heavily than gains [9]. Furthermore, it states that probabilities are perceived subjectively, such that small probabilities are over-estimated for gains and under-estimated for losses [9]. Importantly, 'Prospect Theory' also describes several cognitive biases, i.e., systemic patterns that affect human decision-making and induce violations to the EUT expectation of utility-maximising strategies. Such biases include the 'Endowment Effect', a preference for already-owned objects over non-owned items of the same value [10], or the 'Framing Effect', which explains that humans are risk-seeking when options are presented with positive connotations ('200 of 600 people lived') and risk-averse when they have negative connotations ('400 of 600 people died') [9].

Thereafter, the current consensus in behavioural economics is that human decision-making is a result of a rational utility-maximising strategy, as described by EUT, that interacts with diverse cognitive biases, which are consequences of the social [11], emotional [12], motivational [13], personal [14], and experimental [15] environment of the subject on its economic strategies.

To what extent do human rational utility-maximising strategy and cognitive bias balance compare to animals? What are the evolutionary roots of human decision-making?

There is a good evidence, across species, for utility maximising behaviour, which has led to optimal foraging theory in behavioural ecology ([16, 17]), which states that animal foraging behaviour is maximised in terms of cost-benefit ratio, suggesting that natural selection favours decision-making that leads to economically advantageous foraging behaviour. Indeed, many animal species have very efficient optimisation strategies and appear to be very good at sampling their environment and gaining maximal profit, leading to ideal-free distributions [18]. However, optimal foraging theory is concerned with population-level patterns and not with cognitive mechanisms driving individual decisions, which typically remain unexplored.

The fact that humans often do not make economically optimal choices, especially in risky environments, raises the question of whether the underlying biases may have older, phylogenetically evolved origins before the advent of modern humans. Indeed, previous research with great apes has discovered cognitive biases similar to the 'Framing Effect' (chimpanzees, bonobos [19]) and the 'Endowment Effect' (chimpanzees [20], gorillas [21]) in humans. Further indications for biases and suboptimal decision-making in primates derive from the fact that there can be considerable variability in performance between studies. Indeed, while risk-aversion is frequently described in humans [22] and non-primates [23], risk-preference is harder

to generalise in primates, due to inter-species variability (for instance, bonobos appear more risk-averse than chimpanzees [24–26]) and between-study variance within a given primate species: Rhesus macaques appear risk-prone in [27–29] and risk-averse in [30], chimpanzees appear risk-prone in [31–33] and risk-averse in [34–36], and bonobos appear risk-prone in [19, 32] and risk-averse in [24, 25, 31]. One major issue in this line of research is the diversity of the experimental designs that are used to test primates (see [37] for a review). For instance, a standard and simple design to assess risk-preference is to present a choice between a 'safe' option (providing a fixed and known reward) and a 'risky' option, that may or may not provide a reward (e.g. [24]). In this 'single cup' design, subjects have to *learn* the different possible outcomes of the risky option and the probability to obtain them by memorising outcomes of previous trials. Across species, this experimental design generally (but not always [38]) leads to risk-aversion: bonobos showed 28% of risky choices when both options were equivalent [24], and around 30% of risky choices when the safe option is an intermediately-preferred food reward and the risky option either preferred of non-preferred food reward, with equal probabilities [25]. Chimpanzees showed around 40% of risky choices when the safe option is an intermediately-preferred food reward and the risky option either preferred of non-preferred food reward, with equal probabilities [25], and 34% of risky choices when both options were equivalent [36]. Alternatively, this experimental design can lead to weak risk-proneness: Chimpanzees showed 64% of risky choices when both options were equivalent [24], and chimpanzees and bonobos showed 57% of risky choices when the safe option is an intermediately-preferred food reward and the risky option either preferred of non-preferred food reward, with equal probabilities [39].

More complex designs can be used to assess risk-preference: in [40] for instance, the risky reward was not presented under one risky cup, but in a series of cups, one of which hid the reward (see Fig 1). In this 'multiple cups' design, subjects do not have to *learn* the different outcomes and *infer* their respective probabilities, but they have to *understand* the relationship between the number of risky cups and the probability to obtain the risky reward.

In this design, all four species of great apes were risk-prone, and, compared to the weak risk-proneness or the risk-aversion observed in the 'single cup' design, exhibited extremely high levels of risk-proneness (for bonobos 75% and for chimpanzees 100% of risky choices [40]) when both options were equivalent.

This relationship between experimental design and risk-preference appears to be an indicator of one or more cognitive biases that affect how subjects perceive the safe or the risky option

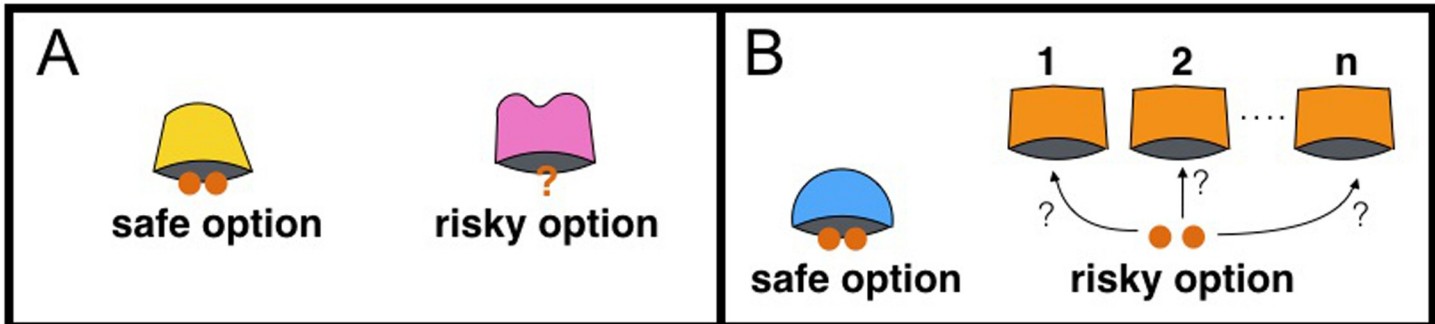

**Fig 1. 'Single cup' design and 'multiple cups' design.** Apparatus for the 'single cup' design (A) as used in [24], leading to risk-aversion or weak risk-proneness. Subjects have to choose between one safe cup and one risky cup yielding an unknown amount of reward, with an unknown probability of yield. Apparatus for the 'multiple cups' design (B) as used in [40], leading to high risk-proneness. Subjects have to choose between one safe cup and one to four risky cups, where only one risky cup contains the reward.

in a given design. If verified, this extreme shift from risk-aversion or weak risk-proneness ('single cup' design) to high risk-proneness ('multiple cups' design) in bonobos and chimpanzees would indicate that, like humans, they are not rational decision-maker (at least in one of the designs) who choose based on expected gain, but that decisions are context-dependant and subjected to cognitive biases.

However, this conclusion remains preliminary for the following reasons. First, in [40], authors kept the quantity of safe rewards smaller than the risky reward, which in itself may have favoured economically irrational decision-making and led subjects to prefer the risky option. Second, subjects were tested with high-quality rewards, which may have introduced a cognitive bias towards risk-taking ([41], [42]). Third, test trials were regularly interspersed with refresher trials (showing subjects the location of the larger, risky reward), which could have introduced a further cognitive bias towards the risky option. Whether or not these design features were sufficient to account for the risk-proneness in [40] remains to be tested. Another concern, raised by the authors themselves, is that subjects might simply have failed to understand the task. In particular, [40] wrote that subjects might be unable '. . .to infer the chances of the risky option without experience and therefore being biased towards a risky choice'.

Indeed, if subjects are tested with variable options, they will inevitably need a number of trials to sample and compute the reward probabilities before being able to take rational decisions. Hence, randomisation of reward probabilities (i.e., changing the number of risky cups) between trials makes it impossible for subjects to learn the probability of the risky option. Instead, apes would have to figure out the relationship between the probability to win and the number of cups by pure inductive reasoning, which may be difficult to do without prior explicit training, even though apes have the cognitive ability to do so ([32], [43], [44]).

In this study, we revisited the question of whether great apes are rational economic decision-makers, by analysing how consistent risk-preference were across the two main experimental designs discussed before ('single cup' vs 'multiple cups' design). We were able to test two species of great apes, gorillas and orang-utans, which have not contributed much to the literature on primate behavioural economics, despite their close phylogenetic relatedness with humans. Indeed, only two studies compared them directly, and they turned out inconsistent: in the study by [40] gorillas and orang-utans were risk-prone; in the study by [45] gorillas and orang-utans were risk-averse, further contributing to the inconsistent patterns already reported in chimpanzees and bonobos. However, a preliminary common conclusion is that orang-utans are less risk-averse than gorillas, which may be linked to differences in feeding, as orang-utans feed on seasonally variable resources [46], using active and costly foraging behaviour [47], while gorillas feed on more steady resources which requires low foraging effort [48], a line of argument already invoked to explain differences between bonobos and chimpanzees [24]. Focusing on these two species (gorillas and orang-utan), in other words, could help clarify various longstanding problems in the behavioural economic literature.

Aside from the species comparison, a second main objective of our study was to determine the impact of experimental design on rational decision-making. In particular, we directly compared two different procedures ('single cup' derived from [24] and 'multiple cups' derived from [40]) with identical reward contingencies, in order to establish whether gorillas and orang-utans made rational choices in both experimental designs. Importantly, we modified the 'multiple cups' design and tested both low- and high-valued reward (as reward value had an impact in earlier studies), added a condition where the safe and risky option were of equal value, avoided refresher trials, counterbalanced the side of the safe and risky options and applied double-blind testing to rule out experimenter bias. To ensure that subjects understood the economic nature of the task, we systematically varied both the amount of reward and the probability to win.

The expectation here is that a rational decision-maker should be relatively more inclined to make a risky choice if the risky option increases or similarly if the probability to win increases. We also tested subjects' rationality by comparing their performance between the two set-ups. The expectation is that a rational decision-maker should have a comparable strategy whatever the set-up, provided that the potential gains and probabilities are identical between the set-ups. However, in line with the literature, we predicted risk-aversion or weak risk-proneness in the 'single cup' design and high risk-proneness in the 'multiple cups' design, hence deviating from rational decision-making.

## Materials and methods

### Subjects

Subjects were five orang-utans (one adult male, three adult females and one juvenile female) and three gorillas (one adult female, one juvenile male and one juvenile female), between 4 and 20 years of age, born in captivity and housed at Basel Zoo, Switzerland, see S1 Table in S1 File. These eight subjects participated in two experiments (experiment 1 and experiment 2 with low-value reward, see Procedure below), and six participated in experiment 2 with high-valued reward, see S1 Table in S1 File. They were kept in a mixed social group (gorillas) or in pairs (orang-utans). Except for one orang-utan, all subjects were naive to any kind of cognition studies.

### Procedure

Experiments were conducted in the indoor part of the enclosures. Subjects were exposed to choice options presented on a trolley that could be moved between enclosures. The trolley was positioned around 30cm from the enclosures. Subjects decided by themselves whether and when they wanted to participate in the trials. Subjects were neither food- nor water-deprived and the regular feeding schedule was continued during data collection. The main research was approved by the cantonal veterinary office of Basel Stadt (permit 2983).

Experiments were conducted during 4-hours morning sessions, five days a week, between February 2018 and February 2020. A maximum of two sessions of 10 trials were conducted every day per subject. The reward we used were 0.5cm$^3$ cubes of food (up to seven pieces at once). Two types of food were used, one low-value food type (vegetables, such as beetroots or carrots, depending on subject preference) and one high-value food type (grapes). All sessions were videotaped, using a HC V500M Panasonic camcorder mounted on a tripod.

Each subject was exposed to one training and two experimental conditions, whose order was counterbalanced between subjects. For the two experiments, the side on the trolley of the safe and the risky options were counterbalanced each day. Finally, we used a double-blind procedure to make sure that the experimenter did not influence the choice of the subjects.

**Training.** In the initial training session, subjects were trained to indicate a cup, remember the content of a cup, and discriminate between different quantities of reward. For this, two transparent little saucers were first placed on the trolley, filled with different quantities of reward. During the first part of the training (i.e., training the subjects to indicate a cup), the experimenter filled one of the saucer with three pieces of food, and the other remained empty. In order to obtain the reward, the subject had to point with their hands (juveniles) or with a stick (adults, as their hand did not fit inside the grid of the enclosure) to the corresponding saucer. After the subject chose a saucer, the experimenter handed the reward (if the chosen cup contained reward) to the subject on a skewer through the mesh of the enclosure.

We collected the pointing stick from the subject's hand after every trial and repositioned it in the enclosure close to the subject at the end of the trial. For each subject, sessions of 10 trials

were run. If, during two consecutive sessions, the subject chose the correct cup in over eight trials out of 10, the first part of the training was completed. During the second part of the training (i.e., training the subjects to remember the content of a cup), the experimental design was similar except that, after filling one of the saucer with three pieces of food, the experimenter hid both saucers with opaque cups (of different shape and colour) in full view of the subject. To complete that part of the training, subjects had to choose the correct cup in over eight trials out of 10, during two consecutive sessions. Finally, during the last part of the training (i.e., training the subjects to discriminate food quantities), the experimenter filled the two saucers with two different quantities of food and covered them with opaque cups (of different shape and colours) in full view of the subject. The two different quantities of food were in that order: 2 vs 0, 2 vs 4, 2 vs 6, 6 vs 7. For one pair of food quantities (ex 2 vs 0) subjects had to choose the correct cup in over eight trials out of 10, during two consecutive sessions, to move on to the next pair of food quantities (ex 2 vs 4). During each part of the training the side of the large and small food quantity was randomised. The duration of the training phase was variable between the subjects (between one and 10 months). Four subjects stopped the experiment during the training phase because of a lack of motivation.

**Experiment 1 –'single cup' design.** In Experiment 1, Experimenter A placed two opaque cups on the trolley, a yellow (safe) cup and a pink (risky) cup, see S1 Fig in S1 File and S1 Movie in S2 File. Next to each cup, she placed a transparent saucer. A wooden occluder was then placed over the risky cup and corresponding saucer. Experimenter A placed 2 pieces of reward (vegetables) in the safe saucer, in full view of the subject. Then she covered that saucer with the safe cup. After that, she either put, or pretended to put, a variable quantity of reward in the risky saucer and covered it with the risky cup, all of which behind the wooden occluder. Then she removed the occluder and left from behind the trolley. Experimenter B, naïve to the condition, then replaced Experimenter A. She waited for the subject to choose one of the cups, revealed the content of both cups, and gave the reward under the chosen cup to the subject (if there was one).

Within the experiment we varied the probability to win and the value of the risky option (see Table 1), while the safe option was fixed. The probability to win corresponded to the probability that a reward was placed under the risky cup (i.e., if P = 0.5, there was one chance out of two that the reward was under the risky cup).

In this design, subjects have to learn the value and the probability of the risky option through the analysis and the memorisation of trials' feedback, so all trials corresponding to the same P*V combinations were performed in a row. For each subject, four consecutive sessions

**Table 1. Expected values (EV) of the risky option across the different conditions of probability to win (P) and of risky reward value (V) in Experiment 1.** The safe option was always set at EV = 2.

|  | P |  |  |  |
|---|---|---|---|---|
| V | 0.25 | 0.33 | 0.5 | 1 |
| 2 | -- | -- | 1 (-) | -- |
| 4 | 1 (-) | 1.33 (-) | 2 (=) | 4 (+) |
| 6 | -- | -- | 3 (+) | -- |

P = probability of obtaining the reward when choosing the risky option; V = value of the reward of the risky option; EV = expected value of the risky option (EV = P*V). The symbol next to the EV indicates whether the risky option has a smaller (-) EV than the safe option, a larger EV (+) than the safe option, or whether the safe and risky option have the same EV (=). Sample sizes: each condition (box) consisted of 40 trials (4 consecutive sessions of 10 trials) per subject. Twenty trials were run per day so each condition took two days to be tested. — = the condition was not tested.

of 10 trials for each of the six combinations of P*V (see the six cells in Table 1) were performed, for a total of 24 sessions (N = 240 trials per subject). The order of the six combinations was randomised between subjects.

**Experiment 2 –'multiple cups' design.** Experimenter A displayed a light-blue (safe) cup and one to four orange (risky) cups on the trolley, and hid the risky cups with the occluder (see S2 Fig in S1 File and S2 Movie in S2 File). Then, she placed two transparent saucers on top of the occluder and filled the 'safe' saucers with two pieces of reward, and one of the 'risky' saucers with two to seven pieces of reward. The safe saucer was then placed under the safe cup in full view of the subject. The risky saucer was placed under one of the risky cups, behind the occluder so that the subject did not know under which cup the saucer was. The position of the baited cup was randomised between trials so that each risky cup held the reward equally often. Experimenter A then removed the occluder and left from behind the trolley. Experimenter B, naïve to the conditions, replaced Experimenter A. She waited for the subject to choose one of the cups, revealed the content of all cups, and gave to the subject the reward that was under the chosen cup (if there was one). Experiment 2 was conducted twice, first using low-valued food reward (vegetables), then after completion, a second time using high-value food type (grapes).

Within the experiment we varied the probability to win (by altering the number of risky cups) and the value of the risky option (see Table 2), while the safe option was fixed.

In this design, subjects have to understand the probability of the risky option through the analysis of the number of risky cups, therefore, to avoid that subjects learn the probability to win over repetition of trials we did not perform consecutively all trials corresponding to the same P*V combination. Each risky reward amount (two, four, six and seven) was tested in two non-consecutive sessions of 10 trials. Among these 20 trials for each risky reward, each cup number (one, two, three or four) was tested five times. The order of the eight sessions (two sessions per risky reward) was randomised between subjects. For each type of reward, the experiment was run twice, to check for learning effects during the experiment, so that each subject was exposed to 16 sessions, i.e., N = 160 trials total (for one type of reward), see S3 Fig in S1 File.

## Statistical analysis

Both experiments were videotaped and subjects' choices (safe or risky option) were coded live during the test. A second observer coded the subjects' choices from the video. Reliability was assessed by recoding 20% of trials which led to excellent reliability (Cohen's k = 0.98). Response times were recorded for each trial (interval between the time when the experimenter

**Table 2. Expected values of the risky option across the different conditions of probability to win (P) and of risky reward value (V) in Experiment 2.** The safe option was always set at EV = 2.

| | P | | | |
|---|---|---|---|---|
| V | 0.25 | 0.33 | 0.5 | 1 |
| 2 | 0.5 (-) | 0.66 (-) | 1 (-) | 2 (=) |
| 4 | 1 (-) | 1.33 (-) | 2 (=) | 4 (+) |
| 6 | 1.5 (-) | 2 (=) | 3 (+) | 6 (+) |
| 7 | 1.75 (-) | 2.33 (+) | 3.5 (+) | 7 (+) |

P = probability of obtaining the reward when choosing the risky option (i.e., 1/number of risky cups); V = value of the reward of the risky option; EV = expected value of the risky option (EV = P*V). The symbol next to the EV indicates whether the risky option has a smaller (-) EV than the safe option, a larger EV (+) than the safe option, or whether the safe and risky option have the same EV (=). Sample sizes: each condition (box) consisted of 10 non-consecutive trials per subject. Twenty trials were run per day.

showed the cups to the subject and the time when the subject pointed unequivocally to one cup).

For our analysis of subjects' choices, we fitted a generalised linear mixed model with binomial error structure and logit link function to our data. The response variable was whether subjects chose the risky option or not. The data were analysed using the glmer function of the lme4 package in R. We checked the normality and the homoscedasticity of plotted residuals and their independence with respect to fitted and other predictors to ensure we met the assumptions of the model. The significance of each predictor variable in explaining variation in rate of risky choices was tested by an analysis of deviance (type II Wald chi-square test).

As our data set originated from a small number of subject, we checked model stability using the influence.ME package in R, that allows to detect influential data in mixed effects models: first, we calculated the estimates of our models when iteratively excluding the influence of each subject (using the function influence of the package), then we computed the Cook's distances measure of every subject on our models (using the cooks.distance function of the package) in order to check whether our models were influenced by certain subjects. If every Cook's distances were inferior to one we concluded that our models were stable.

When analysing risk-preference data in Experiment 1, we fitted the value and the probability of the risky option as continuous variables and the species as a categorical variable. In order to check whether subjects changed their preference throughout the four sessions of each P*V combination, we also fitted the session number as a continuous variable, and the three-way interactions between session, species and probability of the risky option, and between session, species and value of the risky option. In order to test side preference, we fitted the side of the safe cup as a categorical variable. We fitted random intercepts for session within individuals and random slopes of all predictor terms of interest, random slopes of session within individual, and random slopes of all predictor terms of interest within session. We compared this full model to a null model (no fixed effects and the same random structure as the full model) using a Likelihood Ratio Test. If the comparison showed a significant difference, we assessed the significance of each predictor variable in explaining variation in rate of risky choices by an analysis of deviance (type II Wald chi-square test). Then, starting with the highest-level interaction terms, we removed non-significant terms one by one. The final model where all non-significant interaction terms were removed is presented in the result section.

Then, on this final model, we ran post-hoc tests to calculate estimated marginal means or estimated trends (functions emtrends or emmeans of the emmeans package). Finally, in order to check whether the level of risky choice for EV = 2 was significantly different from 50%, we calculated the estimated marginal means for EV = 2 (equivalence point) for each species and each session if session was a significant predictor. If the level of risky choice was not significantly different from 50%, that would indicate risk-neutrality (subjects have no preference between a safe and a risky option of equal expected values). On the contrary, a level of risky choice for EV = 2 significantly higher than 50% would indicate risk-proneness (subjects prefer the risky option), and significantly lower than 50% would indicate risk-aversion (subjects prefer the safe option).

To analyse risk-preference data in Experiment 2 we ran a similar analysis as previously described but we added a predictor variable: the type of reward as a categorical variable. Finally, in order to investigate any positional preference or side bias in Experiment 2, we again fitted a generalised linear mixed model on our data where the response variable was the proportion of selection for each cup (each cup represented by its position on the trolley from left to right, irrespectively of whether it was a risky or a safe cup). We fitted the position of the cup, the total number of cups, the side of the safe cup on the trolley and the species, as well as the two-way interactions between those predictors.

In order to compare the data of Experiment 1 and 2 (with low-valued reward) and to understand the effect of experimental design on risk-preference we ran a similar analysis with a new predictor variable: the experimental design (categorical variable), as well as the two-way interactions between experimental design and economic parameters (probability and value of the risky option). As in the analysis above, we fitted random intercept for session within individuals and random slopes of all predictor terms of interest, random slopes of session within individual, and random slopes of all predictor terms of interest within session, and a random slope of experimental design within individual.

Finally, we analysed the response times in both experiments by fitting a linear mixed-effect model with the response time as the response variable. We log-transformed the response times and fitted a model with the same random structure as before, and with species, economic parameters, type of reward and experimental design as predictors.

## Results

### Experiment 1 ('single cup' design)

In Experiment 1, we varied the probability to win and the value of the risky option (see Table 1, where P refers to the probability of the risky option, V to the value of the risky option, and EV to the expected value of the risky option, while EV = P*V). The results of this experiment are shown in Fig 2 (and S6 Fig in S1 File for individual data). Our full model with all predictors terms and three-way-interactions was significantly different from the null model (LRT: $\chi^2(24) = 138.10$, $p < .001$), see S2 and S3 Tables in S1 File for the random and fixed structure of our models. Table 3 shows our final model where non-significant interaction terms were removed. Cook's distances for the data set of each subject were all smaller than 1; our final model was then evaluated stable across subjects.

Subjects picked the risky cup more often with increasing probability of the risky option ($\chi^2(1) = 90.10$, $p < .001$) and more often with increasing value of the risky option ($\chi^2(1) = 64.72$, $p < .001$). Their level of risky choices was also affected by the session ($\chi^2(3) = 12.45$, $p < .01$) and by the side of the safe cup ($\chi^2(1) = 36.00$, $p < .001$). More importantly, there was a significant interaction between the session and the economic parameters. Firstly, there was a significant interaction between the session and the probability of the risky option ($\chi^2(3) = 12.06$, $p < .01$). Post-hoc tests showed that trend estimates were smaller in sessions 1 and 3 than in session 2 and 4 (see S4 Table in S1 File), i.e., subjects' sensitivity to the probability to win

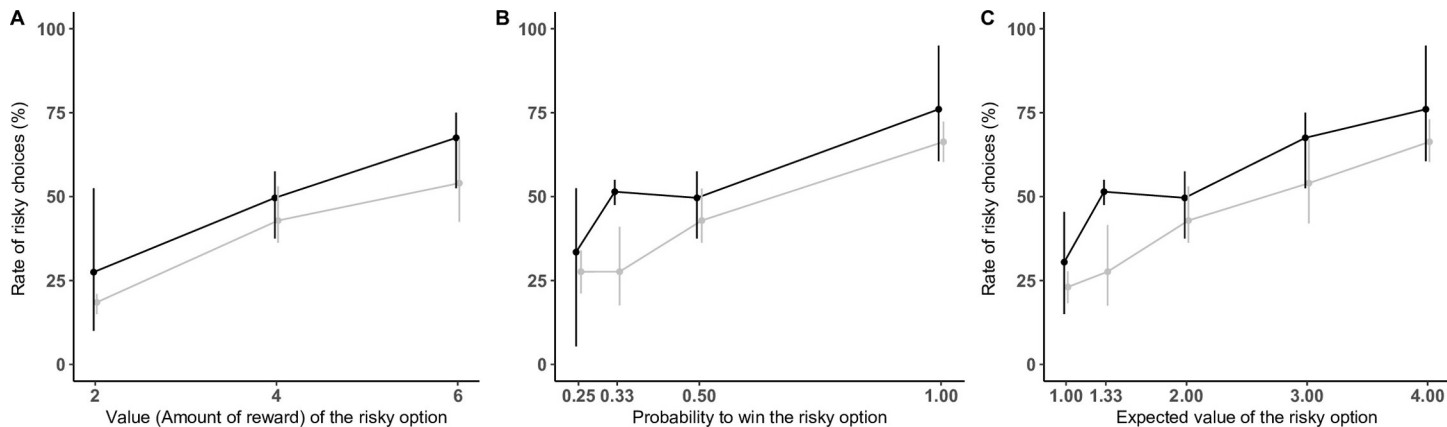

**Fig 2. Results of Experiment 1.** Mean percentage of trials where subjects selected the risky option according to the value of the risky option (A) the probability to win (B) and the expected value of the risky option (C) for gorillas (black) and orang-utans (grey). Error bars indicate 95% confidence intervals.

**Table 3. Fixed effects of the final model investigating subjects' risky preference in Experiment 1.** The table reports the results of the analysis of deviance (type II Wald chi-square tests).

| | Chi-square | Df | p-value |
|---|---|---|---|
| species | 3.21 | 1 | 0.074 |
| side of the safe cup | 36.00 | 1 | < .001 |
| risky probability | 90.10 | 1 | < .001 |
| risky value | 64.72 | 1 | < .001 |
| session | 12.45 | 3 | < .01 |
| species: session | 7.26 | 3 | 0.064 |
| risky probability: session | 12.06 | 3 | < .01 |
| risky value: session | 12.75 | 3 | < .01 |
| species: risky value | 0.03 | 1 | 0.850 |
| species: risky value: session | 12.30 | 3 | < .01 |

increased between the two daily sessions (sessions 1 and 2 being performed the same day, and 3 and 4 the next day). Secondly, there was a significant interaction between session, value of the risky option, and species ($\chi^2(3) = 12.30$, $p < .01$). Post-hoc tests showed that this is mainly because, for gorillas, trend estimates for the value of the risky option differed between session 1 and every other session.

The study of the percentage of risky choices at the indifference point (EV = 2, equality of the expected value of the safe and of the risky option) showed that, for gorillas, the percentage of risky choices when EV = 2 was never significantly different than 50% (interval for estimated marginal means: [0.31;0.64] for session 1, [0.39; 0.73] for session 2, [0.39; 0.72] for session 3, [0.27; 0.60] for session 4). For orang-utans, the percentage of risky choices at EV = 2 was not significantly different from 50% for the two first sessions (interval for estimated marginal means: [0.32; 0.58] for session 1, [0.35; 0.61] for session 2) and was significantly lower than 50% for the last two (interval for estimated marginal means: [0.16; 0.37] for session 3, [0.19; 0.42] for session 4).

## Experiment 2 ('multiple cups' design)

In Experiment 2, we varied the probability to win and the value of the risky option (see Table 2, where P refers to the probability of the risky option, V to the value of the risky option, and EV to the expected value of the risky option, while EV = P*V), as well as the type of reward (low-valued food type: vegetables, and high-valued food type: grapes). The results of this experiment are shown in Fig 3 (and S6 Fig in S1 File for individual data). Our full model with all predictors terms and three-way-interactions was significantly different from the null model (LRT: $\chi^2(25) = 64.94$, $p < .001$), see S5 and S6 Tables in S1 File for the random and fixed structure of our models. Table 4 shows our final model without interactions terms as they were non-significant in the full model.

Cook's distances for the data set of each subject were all smaller than 1, except for one juvenile orang-utan (Ketawa: Cook's D = 1.14, see S6 Fig in S1 File for individual data); our final model was then evaluated relatively stable across subjects.

Subjects picked the risky cup more often with increasing probability of the risky option ($\chi^2(1) = 13.40$, $p < .001$) and more often with increasing value of the risky option ($\chi^2(1) = 47.66$, $p < .001$). Their level of risky choices was also affected by the session ($\chi^2(3) = 10.431$ $p = .01$), by the side of the safe cup ($\chi^2(1) = 11.46$, $p < .001$) and by the reward ($\chi^2(1) = 7.68$, $p < .05$). Investigating the effect of session on subjects' choices, post-hoc tests (see S7 Table in S1 File), showed that the level of risky choices increased throughout session (level of risky

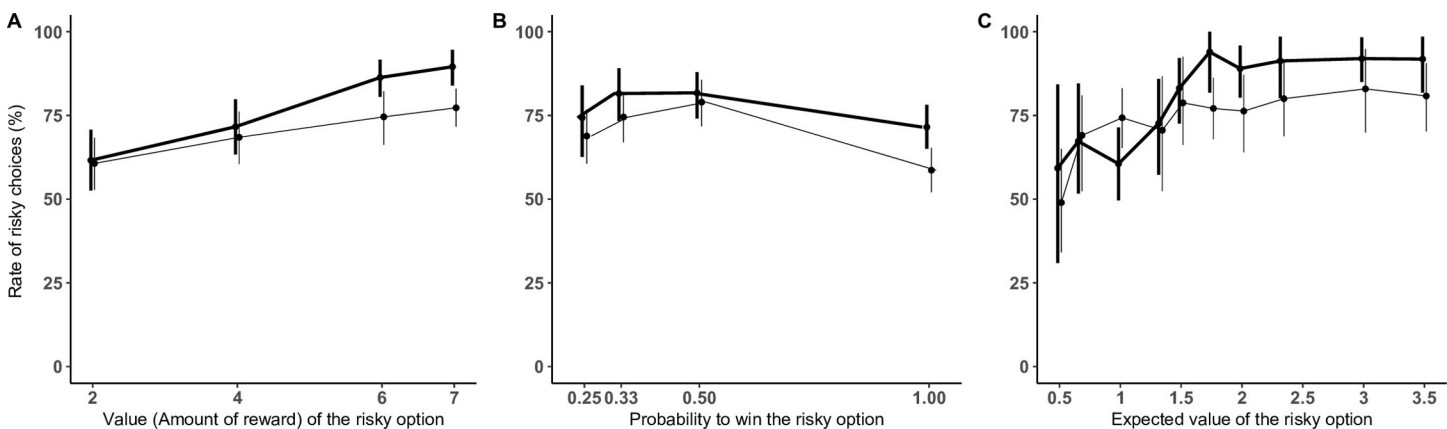

**Fig 3. Results of Experiment 2.** Mean percentage of trials where subjects selected the risky option according to the value of the risky option (A) the probability to win (B) and the expected value of the risky option (C) for the high-valued reward (grapes, in thick line) and low-valued reward (vegetables, in narrow line). Error bars indicate 95% confidence intervals. Only trials with P<1 were considered for (C).

choices ± SE: 0.72 ± 0.07 for session 1, 0.76 ± 0.07 for session 2, 0.80 ± 0.06 for session 3, 0.85 ± 0.05 for session 4). Contrary to what we described in Experiment 1, in Experiment 2 there was no interaction between sessions and risk parameters, which indicates that the effect of session on performance was a general increase of risk-proneness over the experiment and not a learning effect of the probabilities to win over repeated testing.

Finally, our model showed that there was no differences between gorillas and orang-utans ($\chi^2(1) = 1.15$ $p = .28$). The overall percentage of risky choices at the indifference point was 75% (interval for estimated marginal means: [0.63; 0.88]) for low-valued food type) and 81% (interval for estimated marginal means: [0.71; 0.92]) for high-valued food type, i.e., both significantly higher than 50%.

To investigate which parameters influenced the rate of selection of each cup on the trolley (safe and risky cups), we fitted a GLMM model to the percentage of selection of each cup, with the same random structure as before, and with the following predictors: position of the cup on the trolley, side of the safe cup, total number of cups (i.e., risky probability), species. Our full model with all predictors terms and two-way-interactions was significantly different from the null model (LRT: $\chi^2(9) = 54.45$ $p < .001$).

Our final model, see Table 5, showed that percentage of choice of each cup was affected by the overall number of cups ($\chi^2(1) = 52.30$ $p < .001$), and more importantly, that the impact of the position was affected by the total number of risky cups (significant two-way interaction: $\chi^2(1) = 15.65$, $p < .001$), and the species (significant two-way interaction: $\chi^2(1) = 11.40$, $p < .001$). Indeed, see S8 Fig in S1 File, both species showed positional bias: gorillas exhibited a preference for the left location and orang-utans for the central position.

**Table 4. Fixed effects of the final model investigating subjects' risky preference in Experiment 2.** The table reports the results of the analysis of deviance (type II Wald chi-square tests). Only trials with P<1 were considered.

|  | Chi-square | Df | p-value |
|---|---|---|---|
| species | 1.15 | 1 | 0.280 |
| side of the safe cup | 11.46 | 1 | < .001 |
| risky probability | 13.4 | 1 | < .001 |
| risky value | 47.66 | 1 | < .001 |
| reward | 7.68 | 1 | < .05 |
| session | 10.31 | 3 | 0.01 |

**Table 5. Fixed effects of the final model investigating positional biases in Experiment 2.** The table reports the results of the analysis of deviance (type II Wald chi-square tests) for Experiment 2.

|  | Chi-square | Df | p-value |
|---|---|---|---|
| species | 1.83 | 1 | 0.18 |
| side of the safe cup | 0.80 | 1 | 0.37 |
| risky probability | 52.30 | 1 | < .001 |
| position of the cup on the trolley | 1.52 | 1 | 0.22 |
| position of the cup on the trolley: side of the safe cup | 3.06 | 1 | 0.08 |
| position of the cup on the trolley: risky probability | 15.65 | 1 | < .001 |
| position of the cup on the trolley: species | 11.40 | 1 | < .001 |

## Performance in 'single cup' vs 'multiple cups' designs

In order to investigate the impact of the experimental design on subjects' choices, we fitted a full model with the same random structure as the ones we used to analyse Experiment 1 and Experiment 2 with the addition of a random slope of experimental design within individual. We fitted the two economic parameters, the species, the side of the safe cup, the experimental design and the two-way-interactions between experimental design and economic parameters as predictors (see S8 and S9 Tables in S1 File for the random and fixed structure of our models). Our full model was significantly different from the null model (LRT: $\chi^2(7) = 101.10$, $p <$ .001). Table 6 shows our final model where the non-significant interaction terms between risky probability and experimental design was removed. Cook's distances for the data set of each subject were all smaller than 1, except for one juvenile orang-utan (Ketawa: Cook's D = 1.14, see S6 Fig in S1 File for individual data); our final model was then evaluated relatively stable across subjects.

Subjects picked the risky cup more often with increasing probability of the risky option ($\chi^2(1) = 21.43$, $p <$ .001), more often with increasing value of the risky option ($\chi^2(1) = 94.07$, $p <$ .001), and more often in Experiment 2 ($\chi^2(1) = 13.676$ $p <$ .001). More importantly, the interaction between the value of the risky option and the experimental design was significant ($\chi^2(1) = 15.95$, $p <$ .001), but the interaction between the probability to win and the experimental design was not, as we considered only trials with P<1, see Fig 4.

To investigate the impact of the experimental design on subject choices, we ran post-hoc tests (see S10 Table in S1 File) that showed that the level of risky choices was overall higher in Experiment 2 (mean of risky choices for all trials: 78% with a 95% CI: [66%; 90%]) than in Experiment 1 (mean of risky choices for all trials: 39% with a 95% CI: [32%; 47%]), and that the trend estimates for the value of the risky option were steeper in Experiment 1 (trend estimate: 0.1, SE: 0.01) than in Experiment 2 (trend estimate: 0.03, SE: 0.01), see S10 Table in S1 File.

**Table 6. Fixed effects of the final model.** The table reports the results of the analysis of deviance (type II Wald chi-square tests) for Experiment 1 and 2 combined. Only trials with P<1 were considered.

|  | Chi-square | Df | p-value |
|---|---|---|---|
| species | 0.89 | 1 | 0.350 |
| side of the safe cup | 34.35 | 1 | < .001 |
| risky probability | 21.43 | 1 | < .001 |
| risky value | 94.07 | 1 | < .001 |
| experimental design | 13.65 | 1 | < .001 |
| risky value: experimental design | 15.95 | 1 | < .001 |

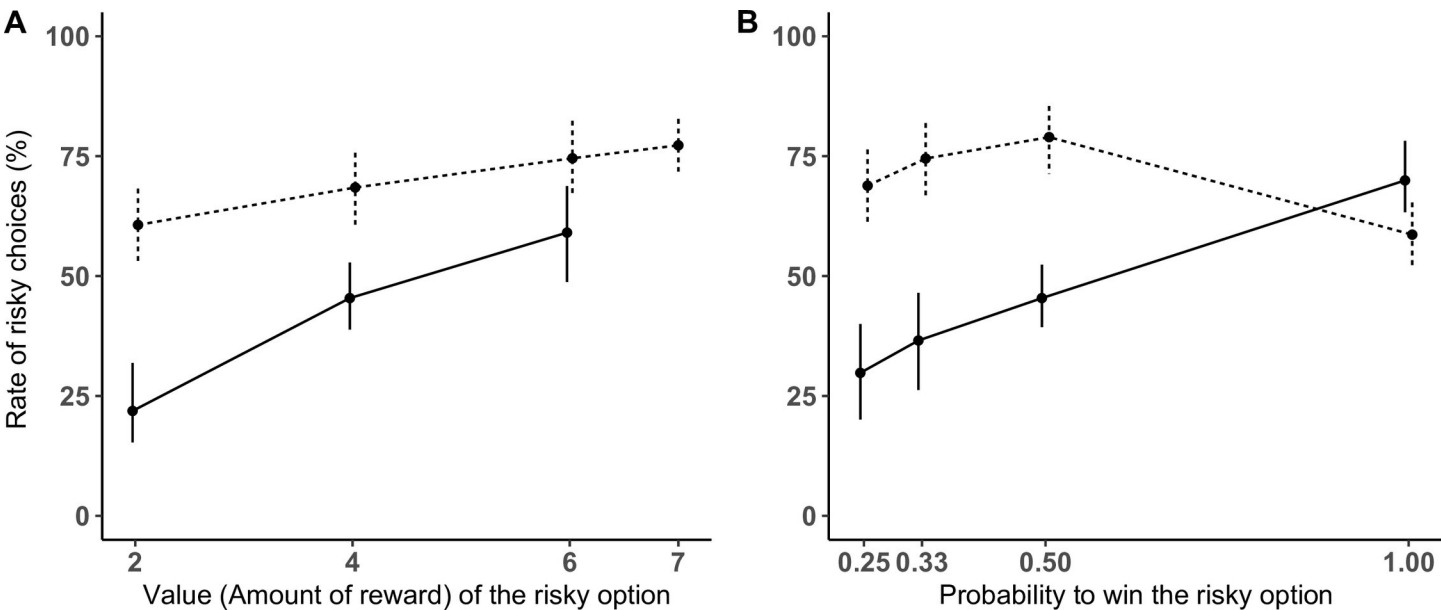

**Fig 4. Performance comparison between experiments.** Mean percentage of trials where subjects selected the risky option for Experiment 1 (solid line) and Experiment 2 (dotted line, low-valued food type): according to the value of the risky option (A) and the probability to win (B). Error bars indicate 95% confidence intervals.

Finally, we analysed the response times in both experiments to assess whether the higher levels of risk-proneness in Experiment 2 could be due to high impulsivity levels, as the risky reward was shown to the subjects before each trial in this design, which could have led subjects to pick the risky option.

We used response time as the response variable after a log-transformation and fitted a model with the same random structure as before, and with species, economic parameters, type of reward and experimental design as predictors. Our model was significantly different from the null model (LRT: $\chi^2(7) = 75.58$, p < .001), and showed that response times were longer in Experiment 2, and longer when the reward was of low-value (respectively $\chi^2(1) = 72.99$, p < .001 and $\chi^2(1) = 8.09$, p < .005), see S9 Fig in S1 File.

## Discussion

We carried out a series of choice experiments in which captive gorillas and orang-utans were asked to choose between two options that differed in risk and reward. In two different experiments, subjects could either choose a safe option that always yielded a predetermined reward or choose a risky option, which could yield a higher reward or no reward at all. In the first experiment, the 'single cup' design, the safe reward was reliably presented under a safe cup, whereas the risky reward was intermittently presented under a risky cup. Here, subjects had to learn from the feedback of previous trials the potential gain and its probability to be hidden under the risky cup. In the second experiment, the 'multiple cups' design, the safe reward was again reliably presented under a safe cup, whereas the risky reward was always presented under one of several risky cups. Here, subjects had to understand the relationship between the number of risky cups and the probability to gain the reward if selecting a risky cup.

We analysed subjects' performance in each of the two experimental designs and found that orang-utans and gorillas generally acted as rational decision-makers, as their preference for the risky option depended on its potential gain and on the probability to win. We found for both experiments that the proportion of risky choices had a linear relationship with the

probability to win, and the amount of reward to win. However, we also found that risk preference was additionally affected by non-economic parameters, as subjects' strategies were not stable across experimental designs. In particular, at the indifference point, subjects were generally risk-neutral in Experiment 1 and risk-prone in Experiment 2. Concerning species comparison, we did not find a performance difference across species in Experiment 2, however, we found that gorillas were more risk-prone than orang-utans in Experiment 1. Our results conflict with previous studies, that reported a species difference in risky context, and concluded that orang-utans are more risk-prone than gorillas ([32, 40]); however, this could be due to the small sample size of our study.

Our major finding, that experimental design affects subjects' risk-preference, is in line with our predictions, as Experiment 1 was derived from an experimental design usually (though not always [38]) leading to risk-aversion ([24, 25, 36]) or weak risk-proneness ([24, 39]) and Experiment 2 was derived from an experimental design that led to high risk-proneness in the four species of great apes [40]. However, the effect of experimental design on subjects' economic strategy had never been directly tested, and the increased risk-proneness triggered by the 'multiple cups' design was so far speculative due to experimental issues, i.e., the safe option was always smaller than the risky option, the side of the safe and risky option were not counterbalanced, visible refreshers trials could have biased subjects toward the risky option [40]. Additionally, authors only used a high-valued reward, which could have increased the levels of risky choices (see [41, 42, 49] for debate on impact of reward on economic decision-making in humans and animals). Finally, [40] could not reject that subjects failed to understand the task and to properly infer the probabilities to win without experience.

In this study, we addressed the experimental issues of the 'multiple cup' design: we used both a low-valued and a high-valued reward, removed refreshers trials, added a condition where the safe and risky option were identical, alternated the side of the safe and risky options. Additionally, we used a double-blind procedure to prevent any kind of experimenter bias. Finally, we investigated how the probability to win affected the level of risky choices (by computing a GLMM with probability to win as a fixed effect) in order to evaluate whether subjects based their strategy on the probability to win and thus whether they were able to assess it. As a result, first, we showed that subjects chose the risky option less often when the probability to win decreased (i.e., when the number of risky cups increased), showing that they understood the economic nature of the task. Second, we found lower levels of risky choices than in [40], indicating that the above-mentioned experimental issues could indeed have artificially increased risk-proneness in the original study.

In sum, our replication of the 'multiple cups' design showed that gorillas and orang-utans understood the economical nature of the task and expressed risk-prone behaviour in this design, even after we removed factors that artificially increased risk-preference in the original design.

As a conclusion, our data confirmed that gorillas and orang-utans were risk-prone in the 'multiple cups' design, while they were risk-neutral in the 'single cup' design. This indicated that, while for each experimental design subjects used a rational strategy and compared potential gain and probabilities to make a decision, apparently irrational contextual factors interacted with this strategy and affected gorillas' and orang-utans' risk-assessment. The irrational finding that the same economic choice (for instance between 'choosing a sure two pieces of carrots' and 'a 50% chance of winning four pieces of carrots') leads to risk-neutrality (no preference) in Experiment 1 and to risk-proneness (preference for the risky choice) in Experiment 2 demonstrated that some aspects inherent in the experimental design of one (or both) experiments triggered one or several cognitive processes affecting how options were perceived.

We put forward several hypothesis accounting for this shift of strategy between the 'single cup' design and the 'multiple cups' design.

A first hypothesis is that this shift was due to a *'description-experience gap'* (D-E gap) as described in [50], which states that decision-makers could experience probability distortions depending on how probabilities are presented to them: either from description (i.e., when probabilities are described to them) or from experience (i.e., they have to sample every option and figure out the probabilities). Indeed, in our study, the 'single cup' design corresponded to an experiment based on experience and the 'multiple cups' design to an experiment based on description, and thus the difference in performance could be explained by such a D-E gap, where subjects were more risk-prone in a design from description ('multiple cups') than from experience ('single cup'). This contradicts previous findings in humans [51] and apes [52] which concluded that designs from experience increased risk-proneness. However, analysing D-E gap is yet subject to debate as several authors question its validity and cause [53], and more importantly, its effect on decision-making seems unclear. Whether the D-E gap induces probabilities distortion for small probabilities (under 20% [54]) or whether it induces risk-preference shift with increased risk-proneness in experiment from experience [51], or the opposite (our findings) remains elusive. Importantly, and contrary to previous studies using visual cues with non-human primates (the probability to gain a reward if choosing a risky option was represented on a screen by the length ratio of coloured bars, for instance a coloured bar divided equally in two colours indicated a 50% probability of winning [52]), we *described* probabilities in a highly intuitive fashion (by a number of risky cups), and avoided mixed *described* designs where subjects could also learn probabilities by sampling. Indeed, in designs that are intermediate between description and experience, all trials with the same rewards contingencies are performed in a row, and subjects have feedback from the chosen option. Finally, both designs provided the same level of feedback (*full feedback*: the content of each option was shown after each trial), while often experiments from experience provide *partial feedback* (only the content of the chosen option is shown after each trial, e.g. [52, 55]), which could lead the decision-maker to pick the risky option solely to gain information, and thus increase the level of risky choices in experiment from experience. In all these respects, this makes our experimental design a highly appropriate one to study D-E gap in non-human primates.

A second hypothesis, the *'impulsivity hypothesis'*, states that higher levels of risky choices in the 'multiple cups' design were due to the fact that the risky reward was shown to the subjects before each trial in this design only. This could have induced an urge to choose that risky reward, regardless of any underlying economic rationality, because of the difficulty to exert self-control over one's impulsive choice ([56, 57] for examples in children, and [58] for an example of how inhibitory control difficulties impact economic choices in mangabeys). Indeed, [40] wrote that '. . .this bias towards the risky option could be explained, for instance, by a failure to inhibit a subject's inherent tendency to choose the large reward, [as] several studies have shown that great apes (and other primates) need a large number of trials to overcome their initial tendency to choose a higher valued food [. . .] even when the reward is no longer visible'. Such a hypothesis could be difficult to test directly, as impulsivity is often measured by proxies such as response times. The analysis of the response times in both experimental designs indicated, on the contrary, longer response times in the 'multiple cups' design than in the 'single cup' design, which do not necessarily reflect differences in impulsivity levels but merely that decision-making in the 'multiple cups' design took longer as there are more cups to choose from and that pointing needs to be more precise, which also takes time. Testing the *'impulsivity hypothesis'* should thus use other means than measuring reaction or responses times, and further studies could try to control subjects' impulsive responses and examine the degree to which the level of impulsivity correlates with risk preference. For instance, previous

work in children [59] showed that if a transparent barrier is placed between subjects and the items they have to pick from (so that they cannot point directly at the desired item but have to make a detour over the barrier to point), automatic and impulsive responses were inhibited. Another study [60] showed similar result when subjects had to wear a weighted bracelet, which also reduced the level of impulsive responses. Such experimental designs (placing transparent barrier between the enclosure grid and the trolley, or providing subjects with weighted sticks,. . .) could be tested in the 'multiple cups' design to investigate if it would lower gorillas and orang-utans observed risk-proneness.

A third hypothesis to explain the risk-preference shift between the 'single cup' and the 'multiple cups' design is the *'exploration hypothesis'* which states that performance is best explained as a bias towards exploring rather than exploiting the risky option, as these two strategies can compete in risk-assessment tasks [61]. In the 'multiple cups' design, subjects were confronted with an array of risky cups, and each one could or could not contain the reward, while in the standard 'single cup' design, only one cup was available. It is therefore conceivable that the 'multiple cups' design was more salient to apes and so triggered more curiosity and exploratory behaviour, even though subjects had *full feedback* on the content of the cups after each trial. In previous studies using a similar 'multiple cups' design, the difference in economic strategies between risk-neutral capuchins and risk-prone young human children [62] were attributed to difference in exploitation/exploration strategies, as capuchins were argued to rely more on an exploitation strategy of safe options while young children rely more on an exploration strategy of uncertain options [63]. However, in our study, we found no support for that *'exploration hypothesis'*, due to the fact that both species exhibited positional biases in Experiment 2: rather than exploring each risky cup, gorillas exhibited a strong preference for the one that was at their left, while orang-utans preferred the central cups.

This leads to a fourth hypothesis, the *'lateral hypothesis'*, stating that a positional physical bias in Experiment 2 could account for the high levels of risky choices in that experiment only. However, as we counterbalanced the sides of the risky and safe option, only the orang-utans central bias (and not the gorillas left preference) could have had an impact on their choices, and artificially increased the level of risky choices (as the central cup is always a risky cup). This would explain why orang-utans showed lower levels of risky choices than gorillas in Experiment 1 but showed comparable to higher rates of risky choices than gorillas in Experiment 2. Importantly, the interaction between a rational economic decision-making and gorillas' and orang-utans' positional bias could explain the inexplicable drop of risky choices to 50% when the probability to win was P = 1, i.e., when there was only one risky cup in Experiment 2. Indeed, gorillas selected preferentially the cup to their left, which corresponded in 50% of the trials to the risky cup, and orang-utans selected indifferently one of the cup as they could not select a central one. This would suggest that when P = 1, subjects were fully driven by their lateral bias. However, the *'lateral hypothesis'* cannot explain the preference shift from risk-neutrality in Experiment 1 to risk-proneness in Experiment 2 for gorillas, so, even if it could explain a portion of orang-utan increased risk-proneness in Experiment 2, it cannot be the sole explanation for our results.

Finally, a last hypothesis is the *'framing hypothesis'*, stating that the main difference between the 'single cup' and the 'multiple cups' design was how subjects framed the safe and risky option. In the 'single cup' design, as the risky option was not necessarily baited with a reward (it was only baited at a certain probability), subjects could have perceived the safe reward as their 'reference' reward quantity, and they either could win (if they picked the risky option and gained the reward) more than that reference quantity, or they could lose it (if they picked the risky option and did not gain anything). On the contrary, in the 'multiple cups' design, the risky reward (equal or larger than the safe reward) was shown before each trial and the risky

option was necessarily baited with a reward (one of the risky cups was always hiding the risky reward); it is therefore possible that subjects have considered that reward quantity as the reference. In that case, choosing the safe reward meant losing a portion of that reference, and choosing the risky option meant either getting that reference quantity or losing it all. In other words, the 'single cup' design could be viewed as a risk-assessment task in the gain domain, and the 'multiple cups' design could be viewed as a risk-assessment task in the loss domain. As the 'Prospect Theory' [8] (and several empirical studies in humans [64] and non-human primates [19]) showed that primate decision-makers are risk-averse in the domain of gain and risk-prone in the domain of loss, the different framing of the safe and risky reward quantities in the 'single cup' design versus 'multiple cups' design could explain subjects preference shift between those two designs: risk-neutrality in the 'single cup' design (risk-aversion/neutrality in the domain of gain) and risk-proneness in the 'multiple cups' design (risk-proneness in the domain of loss).

Whatever the cognitive biases are, our results showed that assessing whether non-human primates are risk-prone or risk-averse is challenging, and that it depends on the experimental design used to answer that question. This is consistent with what has been shown in human research, where slight variations of experimental design led to significant variations of economic preferences [15]. The first research effort, as shown in this manuscript, was to investigate which cognitive biases are triggered by design modifications of the task and led to such drastic variations of economic preferences between 'single cup' and 'multiple cups' design.

The second research effort was to investigate which of the two experimental designs we presented in our study was the more similar to the natural context in which gorillas and orang-utans have to make economical choices and, hence, which one contained higher ecological validity. Gorillas and orang-utans are mostly herbivores and consume preferentially ripe fruits [65], therefore in the wild their economical choices are related to the search and acquisition of those fruits *(risky option)* over the consumption of less preferred food, such as young sprouts, leaves, or even bark (*safe option*). Previous studies investigated the relationship between feeding ecology and risk-preference ([24, 40]), but analysing such relationship is challenging. However, preliminary conclusions indicated that ape species with a higher appetite for ripe fruit (such as chimpanzees [66] or orang-utans [65]) were more risk-prone than ape species who are more incline to switch to less preferred food when fruit production is low (such as bonobos [67] or gorillas [65]). Our study does not allow to conclude similarly, as, maybe due to our small sample size, we did not show species differences between gorillas and orang-utans in Experiment 2, and we showed that gorillas were more risk-prone than orang-utans in Experiment 1. However, we can still use information on species feeding ecology to examine which of the experimental designs corresponded more accurately to the natural context where gorillas and orang-utans have to make economical choices. As mentioned above, gorillas and orang-utans have to choose between a preferred risky option (travel to distant patches to look for ripe fruit, and risk returning empty-handed), or a safe option (consume less preferred food, as leaves or even bark): they have to make economical choices based on their estimation of several parameters of their environment (probability to encounter fruit tree, average production of fruits, direction to go,. . .), and they only have a partial knowledge of the environment. As such, the economical context they have to face is more similar to that of the 'single cup' design, as in the 'multiple cups' design subjects theoretically have a perfect knowledge of the options they have to choose from (probability to gain the reward, amount of reward). However, wild gorillas and orang-utans also have to make a choice between different patches to explore, and not only between a less-preferred option and a variable preferred option, which resembles more to the 'multiple cups' design. Together, gorillas and orang-utans natural economical context appears as a mix of the two designs: a 'multiple cups' design in which each cup could yield

a preferred reward (i.e., where each cup resembles the risky cup of the 'single cup' design), and where subjects would have to evaluate the average quantity of that preferred reward as well as the probability to win each quantity of preferred reward, and where these two parameters (probability and quantity) would be variable over time (i.e., a risky but also *ambiguous* option, which is not at all an aspect we integrated in our task).

As a conclusion, all the above shows that due to the complexity of the economic decisions that gorillas and orang-utans face in the wild, experimental research necessarily uses simplified designs that fail to reproduce that complexity, hence the difficulty to have a clear picture of their economic strategy and preference, and the variability of economical data induced by the variability of experimental designs.

Future work is needed to explore the different potential cognitive biases we laid out in the above hypothesis and test whether one or several could explain our data. Additionally, it would be useful to confirm our findings with a more significant sample size, especially to deeper investigate the species comparison between gorillas and orang-utans. Finally, other primate species (including humans) should be tested with the same designs, as it is currently unclear whether there are systematic species differences in economic decision making. With simple experimental designs, as presented in this study, it should be possible to develop a more comprehensive phylogeny of economic decision making and cognitive biases in human and non-human primates, a further piece in the overall puzzle of the evolutionary origins of intelligence.

## Conclusions

We carried out two experiments to investigate the rationality and consistency of economic decision-making in captive gorillas and orang-utans, in order to fill a gap in the literature on those two species. We chose experimental designs that have already been used in previous research and were suspected to induce different economic strategies, but we added important modifications to rule out lower-level explanations for potential differences in performance between them. Our results show that even though gorillas and orang-utans rely on rational cues (potential gain, probability to win) to establish risk-preference, they are subject to context-based cognitive biases that affect their preference. Indeed, we demonstrated a risk-preference shift from risk-neutrality to risk-proneness depending on the experimental design. The cognitive biases responsible for this shift, such as description-experience gap, impulsivity, positional bias, framing effect, are currently unknown and require targeted research.

## Supporting information

**S1 File. Contains all the supporting tables and figures.**
(PDF)

**S2 File.**
(DOCX)

## Acknowledgments

We are grateful to Basel Zoo for allowing us to conduct research with their great apes. We are thankful to Dr. Radu Slobodeanu for statistical advice.

## Author Contributions

**Conceptualization:** Penelope Lacombe, Christoph D. Dahl.

**Data curation:** Penelope Lacombe, Sarah Brocard.

**Formal analysis:** Penelope Lacombe.

**Funding acquisition:** Klaus Zuberbühler, Christoph D. Dahl.

**Investigation:** Penelope Lacombe, Sarah Brocard.

**Methodology:** Penelope Lacombe, Sarah Brocard.

**Project administration:** Klaus Zuberbühler, Christoph D. Dahl.

**Resources:** Klaus Zuberbühler, Christoph D. Dahl.

**Supervision:** Klaus Zuberbühler, Christoph D. Dahl.

**Writing – original draft:** Penelope Lacombe.

**Writing – review & editing:** Penelope Lacombe, Klaus Zuberbühler, Christoph D. Dahl.

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
