## [Decision Letter · Decision Letter 0]

29 Oct 2021

PONE-D-21-29220Rationality and cognitive bias in captive gorillas and orang-utans economic decision-makingPLOS ONE

Dear Dr. Lacombe,

Thank you for submitting your manuscript to PLOS ONE. After careful consideration, we feel that it has merit but does not fully meet PLOS ONE’s publication criteria as it currently stands. Therefore, we invite you to submit a revised version of the manuscript that addresses the points raised during the review process.

Please carefully address all the points raised by both reviewers, paying particular attention to (i) correctly citing the literature on previous risk preference studies in nonhuman primates (as evidenced by both reviewers, not all nonhuman primate groups tested so far were risk averse; for instance, capuchins tested by the De Petrillo et al 2015 – your reference # 22 – were risk prone when the risky and the safe options had the same EV); (ii) better address in the Discussion the limits of your study in terms of the small sample size, which does not allow to properly evaluate age and sex effects; (iii) take into account all the methodological and statistical concerns raised by Reviewer 1.

Please take also into account the following issues:

Please carefully proof-read the manuscript; I spotted some typos and misspellings.ll 253-55 the graphical solution doesn’t seem ideal, maybe it would be better to use plain text with three different symbols beside each number for indicating whether the EV of the risky option was higher, equal or lower than that of the safe optionl 286 shouldn’t it be EV=2?L 304 “to check for within learning effects”: something seems to be missing hereL 327 “paired t-test”: didn’t you use single-sample t tests to assess whether risky choices significantly differed from the chance level?L 361: ‘Multiple cup’ designL 377 “Results of Experiment 2” is repeated twiceL 432 perhaps “had” rather than “formed” here?L 495 “with feedback from the chosen option”L 560 “do not gain”L 562 please delete “that”L 566 “a” rather than “an”L 596 “species”I suggest to delete, or extensively rephrase, the paragraph on ll 596-615, as it does not point out at the feeding ecology differences between gorillas and orangutans (and, indeed, a thorough discussion on the species differences is not warranted given the small sample sizes) and I am not sure whether it constructively contributes to the Discussion. L 630 please delete “of” (repeated twice)L 636 “on” rather than “towards”?LL 672-3 “respectively”Fig 2 “(B)” is missingSupplementary material: is it the spelling of the video location correct? Shouldn’t it be “zenodo” rather than “zenedo”? Please submit your revised manuscript by Dec 13 2021 11:59PM. If you will need more time than this to complete your revisions, please reply to this message or contact the journal office at plosone@plos.org. Please include the following items when submitting your revised manuscript:A rebuttal letter that responds to each point raised by the academic editor and reviewer(s). You should upload this letter as a separate file labeled 'Response to Reviewers'.A marked-up copy of your manuscript that highlights changes made to the original version. You should upload this as a separate file labeled 'Revised Manuscript with Track Changes'.An unmarked version of your revised paper without tracked changes. You should upload this as a separate file labeled 'Manuscript'.

We look forward to receiving your revised manuscript.

Kind regards,

Elsa Addessi

Academic Editor

PLOS ONE

Journal Requirements:

Reviewers' comments:

Reviewer's Responses to Questions

**Comments to the Author**

1. Is the manuscript technically sound, and do the data support the conclusions?

Reviewer #1: Partly

Reviewer #2: Yes

2. Has the statistical analysis been performed appropriately and rigorously? 

Reviewer #1: N/A

Reviewer #2: Yes

3. Have the authors made all data underlying the findings in their manuscript fully available?

Reviewer #1: Yes

Reviewer #2: Yes

4. Is the manuscript presented in an intelligible fashion and written in standard English?

Reviewer #1: Yes

Reviewer #2: Yes

5. Review Comments to the Author

Reviewer #1: In this study, the authors report two experiments to test rational decision-making in orangutans and gorillas in two risky choice paradigms. The topic is relevant, timely and of interest to the primate decision-making community. It is important to develop a more differentiated view on risk preferences of different species and what modulates them. Adding a sample of Gorillas and Orangutans to the picture is a valuable addition. Having said this, I have three main points of critique that I think need to be addressed carefully.

1. Premise isn’t solid. The authors state “Generally, most primates and non-primates are risk-averse” (l 99). This is contradictory to what they say prior to this (cognitive biases etc, showing context specificity of risk preferences in humans) and, importantly, this generic statement is also not adequate for nonhuman primates as a group. For example, several studies found chimpanzees to be risk-seeking (sometimes in comparison to bonobos) with procedures similar to the two-cup method (e.g., Heilbronner et al., 2008; Rosati & Hare, 2011, 2012, 2013) and in another study the authors cite already, different apes and monkeys were not throughout risk-avoidant (Broihanne et al., 2018). Since the introduction, in its current form, relies on this premise to identify the knowledge gap that the current study aims to fill, it doesn’t seem fit. There are also very different procedures, e.g. eye gaze instead of pointing to a cup, that find nonhuman primates to be risk-seeking sometimes.

In sum, I think the introduction needs some work to adequately reflect the state of current literature and integrate the current study coherently into this picture.

Some examples of relevant literature re risk-proneness in chimpanzees and eye gaze method and two recent studies showing chimps to be more risk-averse, one using a two-cup setup and one using a more complicated apparatus

• Rosati AG, Hare B. 2012 Decision making across social contexts: competition increases preferences for risk in chimpanzees and bonobos. Anim. Behav. 84, 869–879. (doi:10.1016/j.anbehav.2012.07.010

• Rosati AG, Hare B. 2013 Chimpanzees and bonobos exhibit emotional responses to decision outcomes. PLoS ONE 8, e0063058. (doi:10.1371/journal.pone. 0063058)

• Rosati AG, Hare B. 2011 Chimpanzees and bonobos distinguish between risk and ambiguity. Biol. Lett. 7, 15–18. (doi:10.1098/rsbl.2010.0927)

• McCoy, Allison N., and Michael L. Platt. "Risk-sensitive neurons in macaque posterior cingulate cortex." Nature neuroscience 8.9 (2005): 1220-1227.

• Haux, L. M., Engelmann, J. M., Herrmann, E., & Hertwig, R. (2021). How chimpanzees decide in the face of social and nonsocial uncertainty. Animal Behaviour, 173, 177-189.

• Keupp, S., Grueneisen, S., Ludvig, E. A., Warneken, F., & Melis, A. P. (2021). Reduced risk-seeking in chimpanzees in a zero-outcome game. Philosophical Transactions of the Royal Society B, 376(1819), 20190673.

2. I felt that, in general, the results and conclusions should be phrased more carefully, considering the small sample size. Don’t get me wrong, small sample size is not a reason not to publish a study. However, we all have the responsibility to point out obvious limitations in the scope of our analysis and conclusions. Sample sizes of n = 5 and n = 3 neither warrant analysis of individual factors such as sex per species nor statements such as “Our results are in stark contrast to the literature, …” (L 438-440), especially considering that the referred papers also had relatively small sample sizes. The current results add an interesting piece to the puzzle, but I wouldn’t go as far as claiming they show a huge discrepancy, yet. In this respect, I was also wondering if all the individuals showed similar patterns or behaved very differently with perhaps one individual driving the effect? (I raise this point further below in reference to result presentation).

3. Methods & statistics & results: the sections need to be substantially fleshed out. Important pieces of information are missing, the rationale for the statistical approach is not sufficiently explained and model descriptions & results need more detailed information.

Regarding statistics:

I could access the data files, but without accompanying R-code it wasn’t possible to reconstruct which analyses were run. Also, what is a “discontent analysis”? – this is mentioned in the description of the data files but has no match in the manuscript.

The information provided in the manuscript is not sufficient to understand which models were run, which predictors were included, how these predictors were treated (as categorical or continuous) and which model set out to answer which particular question.

For example, l. 309-311 – what does “risky option” refer to (the probability of the option or its size?) and which individual characteristics were assessed?

L. 319-320 – is it correct that ‘session’ entered the model as both fixed and random effect? Also, in none of the models, “protocol” seems to be included as a predictor, but wasn’t this the main goal of the study?

L 318-319: What are the fixed effects? What do you mean with “adding a significant interaction”? If the model is specified as including the interaction term as a predictor, then running the model and comparing it to respective null model will reveal which terms explain considerable parts of the variance. But you can’t only add a predictor term after you found it somehow to be significant – maybe this is a misunderstanding due to the way the sentence is phrased in the current version of the manuscript. But similar wording is in l 335-336, so I wonder what exactly were the steps that the authors performed during their analysis?

Currently, the analysis section reads like a long list of GLMMs and t-tests, but it’s difficult to follow which question each of them answers and which part of the results section refers to which of these models/t-tests. Why do you run additional t-tests when you already test for risk-preference with the GLMMs? In addition, from the information that is provided, random slopes are missing from the GLMMs. However, these are important to model effects of fixed effects among levels of random effects and to avoid inflated Type 1 error rates.

I would like to see some more information on the paired-samples t-Test that was performed to test for side bias (as stated in L 334) and provided the results reported in L 383-390 – I am struggling to see how a t-Test can account simultaneously for species differences, different number of possible positions to choose depending on number of presented cups, and side of safe reward. Or were several t-Tests run on the same data to test for these effects? In this case, which method was used to account for multiple testing, and was there a reason why these aspects were not considered as predictor terms in the GLMM in the first place?

I didn’t find information about inter-rater reliability.

Results:

It was difficult to follow the results description and judge its appropriateness, given the missing information as outlined above. A table specifying the model output would help a lot, specifying for all predictor terms the respective estimates, standard errors, confidence intervals, and test results (likelihood ratio test, degrees of freedom, p-value). And information if the model was overall different from a model excluding the predictor terms of interest; consider including effect size in result reporting, as well.

L 369: this is the first time that is mentioned that not all subjects were tested with all food rewards.

L 355: is this 45.7% for Gorillas or Orangutans? And the other species?

I don’t think assessing effects of age and sex statistically is warranted, given the samples only have 1 male per species, only 1 adult for the orangs and only 1 juvenile for the Gorillas.

L 400-406: which data and which model corresponds to these results?

Figures: It would be nice to see the individual data, especially because there are so few subjects. X-axis labelling should be adapted so it shows the possible values for the different figures (e.g., A only allows values of 2,4,6 and 7 but shows also integers in between, whereas C allows multiple values but only shows 1,2,3, and 4). Confidence intervals overlap for different lines and hide the CI of the other line (species or reward value); applying a jitter function might help.

Caption Fig.2: missing (B). Delete “for low valued reward” – its not necessary because no high value reward condition was presented.

Caption Fig.3: delete “Results of Experiment 2”.

Methods:

L 189-192: please, provide more details about subjects here (at the very least that only 1 male was in each group) and refer to supplementary table.

In the discussion, a pilot study is mentioned; which is contradictory to subjects being completely naïve to testing prior to procedure of Exp.1. Was the pilot part of this study and what was it about? Is it of relevance to the subjects’ testing history?

L 215-28: which quantities were used? What does “success rate of >=80%” mean? Did they have to reach this criterion for different quantity discriminations, or overall, or in a specified number of consecutive test sessions? How many trials did it take the individuals to reach this criterion? These informations are important to get an idea how stable the 80% performance rate was or whether it might have been a lucky accident.

L224-226: Did the subjects first see where food was placed and then pointed to the now hidden rewards, or did they have to learn to associate a cup colour with a specific content by sampling information?

L232-242: did subjects see in advance which quantity might be hidden under the risky cup? Why did you use two experimenters? On the videos it looks like E2 is watching E1 hiding the food, was this always the case?

L260-265: How was number trials decided? (for example, based on a simulation to find out necessary number of trials to find an effect, if there is one; or based on previous literature)

Design table: in a 10-trial session, how were the different win probabilities realized? How many wins and losses were presented per session per condition? It doesn’t add up for me. For example, .33 -> did you bait 3 or 4 wins within 10 trial session?

Exp. 2: did all subjects of Exp1. Participate in Exp. 2? Was there a break between the experiments?

Did they receive a familiarization training with the new procedure to learn that even when there are more cups than previously, they still only get to pick 1 and not more?

Why was the order of food type not counterbalanced between individuals?

Why is there an additional value (7 rewards) for Exp 2?

L297-299: I don’t understand this reasoning. Why was it necessary to prevent subjects from learning about the win probability? Isn’t it necessary for the subject to understand the probabilities to make an informed decision about whether to play it safe or not?

L301-303: this sentence is confusing – how is each reward amount, all numbers of cups be tested 5 times within a session that only contains 10 trials?

L297-305: Its very hard to follow the description of number of trials, sessions, and experiment repetitions. E.g. l303-> could you just say that you presented 16 sessions per reward type, i.e. 32 sessions in total? It might help to present an example of one of the orders (either in the main text or as part of the supplementary material).

I was missing an explanation why not all expected values tested in Exp.1. And why not both high and low quality food types? Seems to make it difficult to directly compare Exp. 1 and Exp.2.

Discussion

The authors raise interesting points and embed the results well, raising several possibilities for the differences they found between Exp. 1 and Exp. 2. Despite the discussion being already in a good shape, I have some remaining questions and remarks.

L 443-444: “an experimental design usually leading to risk-aversion [26]”. But Ref 26 found risk aversion in bonbos and risk proneness in chimps and thus doesn’t fit the statement very well

L 452-454: “Finally, [28] could not reject that subjects understood the task and properly inferred the probabilities to win without experience. In this study, we fixed these experimental issues” -> explanation required how exactly the current study fixed the issue.

L 456-457: On the videos provided, it looks like the baiting procedure is well in sight of Experimenter 2. Was this always the case?

L 457: “levels of risky choices” -> does this refer to the size of the risky option?

L 458-459: “probability to win to verify that subjects based their strategy on the value of that probability and were thus able to compute it” -> its not fully clear to which of the listed results this is referring.

L 512. What pilot study? It is mentioned the first time here, should the reader know about it already at this point?

The “lateral hypothesis”. The position bias is interesting, and also that the two species apparently had different kind of bias. However, what does this tell us about the risky choice results? Are they meaningful at all when subjects had a clear position bias? I think this point deserves mention and being discussed.

Reviewer #2: The article aims to investigate how gorillas and orangutans behave in two risk-assessment tasks. In experiment 1, the apes experienced a single-choice scenario where they had to decide between a cup containing a secure reward and a cup that may contain a high-value reward. Importantly, they did not know beforehand the quantity of reward at stake and whether a reward would be present or not at any given trial—although they might learn the probability that the reward was present after some experience. In experiment 2, apes experienced a multiple-choice scenario where again they had to decide between a cup containing a secure reward and a choice between a maximum of four risk options—only one being baited with a reward. Importantly, in the second study, the apes knew the risky reward at stakes and that it was always baited inside one of the cups. They found that both species seem to act rationally, increasing their choices towards the risk option when the expected value for the risk option is higher, when the probability to obtain it is higher and when its value is higher. However, they find differences between experiments, with both species being more risk-prone in experiment 2 (in those trials in which the expected value of the secure and the risky choice was the same) and differences between species, with gorillas being more risk-prone than orangutans in experiment 1.

The studies are well-conducted, and the statistical analysis is correct. In the discussion, the authors do a good job describing potential hypotheses to explain their results. However, I would like the authors to clarify some points before final acceptance.

Uncertainty vs. risk tasks

My main comment concerns the difference between uncertainty/ambiguity and risk scenarios. The authors interpret their two studies as tasks suitable to test risk preferences. However, in their discussion, they comment that during the single-cup experiment 1, apes learn the probabilities by experience (as opposed to experiment 2, in which the exact probabilities are described since apes can observe the risky reward and the number of cups). While I agree with the distinction, I am surprised that the authors do not discuss differences in terms of ambiguity/risk. One possibility is that in experiment 1, it is harder to learn the probability of appearance despite the experience. Although the study reports that a rise in the probability to win had an increasing effect on apes risk choices, one could imagine that apes were often deciding under conditions of ambiguity (especially at the beginning of experiment 1 session), and that could partially explain differences between experiments 1 and 2 overall preferences towards risk. If that were the case, the results would also align with previous studies by Rosati & Hare, 2011 and Haux et al., 2021.

Introduction

L 99 and 105: I am not sure if the ref. 26 interpretation is accurate. The chimpanzees were risk-prone in that study. Furthermore, risk options always provided rewards (either less or more grapes than the secure option).

L 132: I would stick with rewards instead of awards.

L 180: Check if the hypothesis are correct. The authors mention that they expect risk proneness in both tasks, but in L 99, they argue that primates are mostly risk-averse. I guess the authors expect more risk aversion or neutrality in experiment 1, as they have found.

Training:

In general, some details are missing. Which were the quantities involved in the training? Did they need to reach a rate of 80% correct over how many trials?

Also, what was the purpose of the hidden condition, to demonstrate that they can remember the quantities through the transparent saucer before they are covered? Did they then also need to reach at least an 80% of success in those trials? Please specify this in the manuscript.

Experiment 1:

L 260: I wonder if the authors could analyze the session effect in experiment 1 as they did in experiment 2 (instead of having session as a random effect). A session effect could tell us whether apes were learning the EV across time—increasing or decreasing their risk choices depending on the probability of obtaining the reward (P) and its value (V). The last comment relates to my previous one on the difference between ambiguity/risk choices.

It might also help to specify whether the presentation order varied between the four sessions. For instance, if the probability of food present in the risk choice was 0.5, was food present in the risk option every second trial, or was the presentation randomized as long as there would be a total of 5 trials with and without food? If trial presentation order was blocked within sessions, then ambiguity was reduced since the four sessions would be exactly the same.

Statistical analysis:

The description of the models is very clear except for the fixed-effects part. The authors describe the random effects and the additional fixed effects for experiment 2, but it is unclear which are the shared fixed effects between studies (e.g., 318). It is only apparent in the result section (e.g., the value of the risky option).

L 323: another instead of an other.

Results:

I would remind the readers that the probability of the risky option refers to P in the table, that the value of the risky option refers to V and that the expected value refers to EV =P*V.

Discussion:

L 444: Ref 26 leads to risk aversion only in bonobos.

L 487: Close parenthesis after reference 43.

L 492: either "described" or 'described'.

L 493 to 495: I find these sentences slightly unclear (e.g., "with feedback the chosen option").

L 512: It might be interesting to report some descriptives of the pilot regarding the latencies discussed.

L 563: I understand the logic in the possible distinction between the risk assessment task in the domain of gains (experiment 1) and the domain of losses (experiment 2). However, how do we know whether apes took the risk choice as the reference in experiment 2? Perhaps the authors could cite some prospect theory literature showing whether better options are usually taken as reference points?

Also, the authors mention that, given that the risk option is the reference, choosing it leads to the loss of all the reference quantity—instead of a proportion of it when choosing the safe option. However, that might not be true for all trial constellations since sometimes the risk option is equally probable, and they can also obtain it from time to time. In other words, they do not permanently lose all the “reference” quantity. Otherwise, they would always take the safe option.

L 596: close parenthesis after Experiment 2.

L 610: I would suggest making the potential role of single cup design more explicit.

6. PLOS authors have the option to publish the peer review history of their article (what does this mean?). If published, this will include your full peer review and any attached files.

Reviewer #1: No

Reviewer #2: No

---

## [Author Response · Author response to Decision Letter 0]

13 Dec 2021

Cover letter and detailed answers to the editor’s and the reviewers’ comments

Ref.: Ms. No. PONE-D-21-29220

Rationality and cognitive bias in captive gorillas and orang-utans economic decision-making

PLOS ONE

Dear Dr Elsa Addessi, Academic Editor of PLOS ONE,

We would like to express our appreciation for your and the reviewers' useful comments and contribution, especially concerning the literature and the methodological aspects (figures and statistical analyses). We implemented all changes as suggested by the editor and the reviewers. In the following we addressed all comments and questions in details.

Our best regards,

Pénélope Lacombe, Christoph Dahl.

Editor:

correctly citing the literature on previous risk preference studies in nonhuman primates (as evidenced by both reviewers, not all nonhuman primate groups tested so far were risk averse; for instance, capuchins tested by the De Petrillo et al 2015 – your reference # 22 – were risk prone when the risky and the safe options had the same EV); 

Modification of the manuscript : lines 95 to 119.

better address in the Discussion the limits of your study in terms of the small sample size, which does not allow to properly evaluate age and sex effects;

We modified the manuscript following this suggestion (the effect of age and sex are no longer analysed) and we addressed the limitation of our conclusions (lines 677 and 706).

 take into account all the methodological and statistical concerns raised by Reviewer

Modification of the manuscript : lines 326 to 369.

253-55 the graphical solution doesn’t seem ideal, maybe it would be better to use plain text with three different symbols beside each number for indicating whether the EV of the risky option was higher, equal or lower than that of the safe option

Modification of the manuscript : table 1 and 2. 

l 286 shouldn’t it be EV=2?

Modification of the manuscript : line 305.

L 304 “to check for within learning effects”: something seems to be missing here

Modification of the manuscript : line 322.

L 327 “paired t-test”: didn’t you use single-sample t tests to assess whether risky choices significantly differed from the chance level?

Indeed, this was an error in the manuscript. Modification of the manuscript : line 370.

L 361: ‘Multiple cup’ design

Modification of the manuscript : line 415.

L 377 “Results of Experiment 2” is repeated twice

Modification of the manuscript : line 437.

L 432 perhaps “had” rather than “formed” here?

Modification of the manuscript : line 513.

L 495 “with feedback from the chosen option”

Modification of the manuscript : line 579.

L 560 “do not gain”

Modification of the manuscript : line 644.

L 562 please delete “that”

Modification of the manuscript : line 646.

L 566 “a” rather than “an”

Modification of the manuscript : line 650.

L 596 “species”

Modification of the manuscript : line 677.

I suggest to delete, or extensively rephrase, the paragraph on ll 596-615, as it does not point out at the feeding ecology differences between gorillas and orangutans (and, indeed, a thorough discussion on the species differences is not warranted given the small sample sizes) and I am not sure whether it constructively contributes to the Discussion. 

The aim of this paragraph was not to discuss species differences but to discuss the relevance of each protocol in the study of economic strategy in great apes. Our point was that, in order to investigate great apes’ economic decision-making, one should confront the subjects with similar choices as the ones they would face in the wild. The conclusion is that, given the ecology of gorillas and orang-utans, a mixed of both designs would have more ecological relevance.

L 630 please delete “of” (repeated twice)

Modification of the manuscript : line 712.

L 636 “on” rather than “towards”?

Modification of the manuscript : line 718.

LL 672-3 “respectively”

Modification of the manuscript : lines 761, 762.

Fig 2 “(B)” is missing

Modification of the manuscript : Fig 2.

Supplementary material: is it the spelling of the video location correct? Shouldn’t it be “zenodo” rather than “zenedo”?

Modification of the manuscript :lines 769, 772.

Reviewer #1: 

Premise isn’t solid. The authors state “Generally, most primates and non-primates are risk-averse” (l 99). This is contradictory to what they say prior to this (cognitive biases etc, showing context specificity of risk preferences in humans) and, importantly, this generic statement is also not adequate for nonhuman primates as a group. For example, several studies found chimpanzees to be risk-seeking (sometimes in comparison to bonobos) with procedures similar to the two-cup method (e.g., Heilbronner et al., 2008; Rosati & Hare, 2011, 2012, 2013) and in another study the authors cite already, different apes and monkeys were not throughout risk-avoidant (Broihanne et al., 2018). Since the introduction, in its current form, relies on this premise to identify the knowledge gap that the current study aims to fill, it doesn’t seem fit. There are also very different procedures, e.g. eye gaze instead of pointing to a cup, that find nonhuman primates to be risk-seeking sometimes.

In sum, I think the introduction needs some work to adequately reflect the state of current literature and integrate the current study coherently into this picture.

Some examples of relevant literature re risk-proneness in chimpanzees and eye gaze method and two recent studies showing chimps to be more risk-averse, one using a two-cup setup and one using a more complicated apparatus

Modification of the manuscript : lines 95 to 119.

I felt that, in general, the results and conclusions should be phrased more carefully, considering the small sample size. Don’t get me wrong, small sample size is not a reason not to publish a study. However, we all have the responsibility to point out obvious limitations in the scope of our analysis and conclusions. Sample sizes of n = 5 and n = 3 neither warrant analysis of individual factors such as sex per species nor statements such as “Our results are in stark contrast to the literature, …” (L 438-440), especially considering that the referred papers also had relatively small sample sizes. The current results add an interesting piece to the puzzle, but I wouldn’t go as far as claiming they show a huge discrepancy, yet. In this respect, I was also wondering if all the individuals showed similar patterns or behaved very differently with perhaps one individual driving the effect? (I raise this point further below in reference to result presentation).

We modified the manuscript following this suggestion (the effect of age and sex are no longer analysed) and we addressed the limitation of our conclusions (lines 677 and 706). For individual data, see below.

Methods & statistics & results: the sections need to be substantially fleshed out. Important pieces of information are missing, the rationale for the statistical approach is not sufficiently explained and model descriptions & results need more detailed information.

Modification of the manuscript : lines 326 to 369.

Also, what is a “discontent analysis”? – this is mentioned in the description of the data files but has no match in the manuscript.

We removed that analysis of our study.

The information provided in the manuscript is not sufficient to understand which models were run, which predictors were included, how these predictors were treated (as categorical or continuous) and which model set out to answer which particular question.

For example, l. 309-311 – what does “risky option” refer to (the probability of the option or its size?) and which individual characteristics were assessed?

L. 319-320 – is it correct that ‘session’ entered the model as both fixed and random effect? Also, in none of the models, “protocol” seems to be included as a predictor, but wasn’t this the main goal of the study?

L 318-319: What are the fixed effects? What do you mean with “adding a significant interaction”? If the model is specified as including the interaction term as a predictor, then running the model and comparing it to respective null model will reveal which terms explain considerable parts of the variance. But you can’t only add a predictor term after you found it somehow to be significant – maybe this is a misunderstanding due to the way the sentence is phrased in the current version of the manuscript. But similar wording is in l 335-336, so I wonder what exactly were the steps that the authors performed during their analysis?

Currently, the analysis section reads like a long list of GLMMs and t-tests, but it’s difficult to follow which question each of them answers and which part of the results section refers to which of these models/t-tests. Why do you run additional t-tests when you already test for risk-preference with the GLMMs? In addition, from the information that is provided, random slopes are missing from the GLMMs. However, these are important to model effects of fixed effects among levels of random effects and to avoid inflated Type 1 error rates.I would like to see some more information on the paired-samples t-Test that was performed to test for side bias (as stated in L 334) and provided the results reported in L 383-390 – I am struggling to see how a t-Test can account simultaneously for species differences, different number of possible positions to choose depending on number of presented cups, and side of safe reward. Or were several t-Tests run on the same data to test for these effects? In this case, which method was used to account for multiple testing, and was there a reason why these aspects were not considered as predictor terms in the GLMM in the first place?

Modification of the manuscript : lines 326 to 369.

I didn’t find information about inter-rater reliability.

Modification of the manuscript : lines 326 to 328.

Results:

It was difficult to follow the results description and judge its appropriateness, given the missing information as outlined above. A table specifying the model output would help a lot, specifying for all predictor terms the respective estimates, standard errors, confidence intervals, and test results (likelihood ratio test, degrees of freedom, p-value). And information if the model was overall different from a model excluding the predictor terms of interest; consider including effect size in result reporting, as well.

Modification of the manuscript : tables 3, 4, 5 and 6. The result sections were also re-written following your suggestions (lines 382 to 397, 416 to 432, 452 to 455, and 473 to 475).

L 355: is this 45.7% for Gorillas or Orangutans? And the other species?

Modification of the manuscript : lines 400 to 403.

I don’t think assessing effects of age and sex statistically is warranted, given the samples only have 1 male per species, only 1 adult for the orangs and only 1 juvenile for the Gorillas.

We modified the manuscript following this suggestion (the effect of age and sex are no longer analysed).

L 400-406: which data and which model corresponds to these results?

Modification of the manuscript : table 6.

Figures: It would be nice to see the individual data, especially because there are so few subjects. 

Individual data for Experiment 1

Individual data for Experiment 2 (low-level reward)

X-axis labelling should be adapted so it shows the possible values for the different figures (e.g., A only allows values of 2,4,6 and 7 but shows also integers in between, whereas C allows multiple values but only shows 1,2,3, and 4). Confidence intervals overlap for different lines and hide the CI of the other line (species or reward value); applying a jitter function might help.

Modification of the manuscript : figures 2, 3, 4.

Caption Fig.2: missing (B). Delete “for low valued reward” – its not necessary because no high value reward condition was presented.

Modification of the manuscript : figure 2.

Caption Fig.3: delete “Results of Experiment 2”.

Modification of the manuscript : figure 3.

Methods:

L 189-192: please, provide more details about subjects here (at the very least that only 1 male was in each group) and refer to supplementary table.

Modification of the manuscript : lines 195 to 200. 

In the discussion, a pilot study is mentioned; which is contradictory to subjects being completely naïve to testing prior to procedure of Exp.1. Was the pilot part of this study and what was it about? Is it of relevance to the subjects’ testing history?

The wording “pilot study” was confusing. We just analysed the response times in both experiments and received the following results (see figure below).

L 215-28: which quantities were used? What does “success rate of >=80%” mean? Did they have to reach this criterion for different quantity discriminations, or overall, or in a specified number of consecutive test sessions? How many trials did it take the individuals to reach this criterion? These informations are important to get an idea how stable the 80% performance rate was or whether it might have been a lucky accident.

Modification of the manuscript : lines 234 to 247.

L224-226: Did the subjects first see where food was placed and then pointed to the now hidden rewards, or did they have to learn to associate a cup colour with a specific content by sampling information?

Modification of the manuscript : lines 234 to 247.

L232-242: did subjects see in advance which quantity might be hidden under the risky cup?

 No, they did not. 

Why did you use two experimenters? 

To conduct a double-blind procedure.

On the videos it looks like E2 is watching E1 hiding the food, was this always the case? 

No, this was not the case: E2 stayed close to E1 but was not looking at where the food was hidden. 

L260-265: How was number trials decided? (for example, based on a simulation to find out necessary number of trials to find an effect, if there is one; or based on previous literature).

In Heilbronner et al (2008), subjects were tested in an analogous design of Experiment 1 and the learning effect was analyzed: subjects risk-preference were stable between the 3 blocks of 3 sessions (10 trials per session). 30 trials were then sufficient to establish risk-preference. In Hayden et al (2008), and Long et al (2009) similar tasks (risk-assessment in a design similar to the “single-cup” design) were conducted using blocks of 25 to 40 trials per condition.

As we conducted 20 trials per day, we performed 40 trials (2 sessions of 10 trials) for each PxV combination. 

Design table: in a 10-trial session, how were the different win probabilities realized? How many wins and losses were presented per session per condition? It doesn’t add up for me. For example, .33 -> did you bait 3 or 4 wins within 10 trial session?

For P=0.3, we assigned 1 (baited cup) in every 3 trials, ex : 0 1 0 0 0 1 0 1 0 1 0 0 0 1 0 1 0 0 1 (0 0). The last two trials were not performed as we conducted 20 trials per day. 

For P=0.25, we assigned 1 (baited cup) in every 4 trials, ex : 0 1 0 0 0 0 1 0 etc. 

For P=5, we assigned 1 (baited cup) in every 2 trials, ex : 0 1 0 1 1 0 0 11 0 0 1 etc. 

Exp. 2: did all subjects of Exp1. Participate in Exp. 2? Was there a break between the experiments?

No, one subject (female adult gorilla: Adira) dropped the experiment between E1 and E2, and one subject (male adult orang-utan: Bagus) only did E2+ (E2 with high-reward food), see table S1.These two subjects were removed from the whole experiment. Two subjects that completed E1 only completed E2- (E2 with low-reward food). They were included in the results of E1 and E2- and in the comparison of E1 and E2- (as both experiments used low-reward food), but not in the results of E2+.

There was a break between the experiments of one week. 

Did they receive a familiarisation training with the new procedure to learn that even when there are more cups than previously, they still only get to pick 1 and not more?

No, but they did not try to pick several cups. 

Why was the order of food type not counterbalanced between individuals?

This would have been preferable, but we obtained the authorisation from the veterinary to use high-valued food after we started the experiment. For one subject (Bagus), we started E2 with high-valued food, but he did not performed E2 with low-valued food so he was removed from the experiment. 

Why is there an additional value (7 rewards) for Exp 2?

We wanted to increase the number of conditions where the expected value of the risky option was generally higher than the expected value of the safe option (without the value of 7 rewards we would have tested only one such condition where P<1 and EV > 2). This was important especially for Exp 2 as the design of Exp 1 has been studied and tested on several species before, and previous work showed that subjects understood the economic nature of the task, while Exp 2 has only been performed once (in the same design that we run it) and, prior to our study, it was still unclear weather subjects understood the task.

We were allowed a maximum value of 7 rewards per trial by the zoo veterinarian. 

L297-299: I don’t understand this reasoning. Why was it necessary to prevent subjects from learning about the win probability? Isn’t it necessary for the subject to understand the probabilities to make an informed decision about whether to play it safe or not?

In E2 we did not want subjects to learn the probabilities to win, but to understand and infer probabilities without training from the number of risky cups. This is why the number of risky cups was semi-randomised within a session. 

L301-303: this sentence is confusing – how is each reward amount, all numbers of cups be tested 5 times within a session that only contains 10 trials?

Each reward amount is tested during 2 non-consecutive sessions of 10 trials: among these 20 trials, each cup number (1, 2, 3 or 4) is tested 5 times. 

L297-305: Its very hard to follow the description of number of trials, sessions, and experiment repetitions. E.g. l303-> could you just say that you presented 16 sessions per reward type, i.e. 32 sessions in total? It might help to present an example of one of the orders (either in the main text or as part of the supplementary material).

For one subject and one type of reward:

We performed two runs of the experiment to check for learning effects. For each run of Exp 2, we conducted 16 sessions. During each sessions we tested one reward quantity, and varied the number of risky cups between trials. Throughout the 16 sessions of one run, each PxV combination was tested 5 times (for instance in the table the grey cells corresponds to the 5 occurrences of the V=2, P=0.25 combination for the first run).

Modification of the manuscript : figure S3.

I was missing an explanation why not all expected values tested in Exp.1. And why not both high and low quality food types? Seems to make it difficult to directly compare Exp. 1 and Exp.2.

It would have been preferable to test all values in Exp.1 and to test both quality food types. Unfortunately, the experiment was already quite complex and very long to run (2.5 years), therefore we could not test the combinations of every parameters (EV, type of food) in Experiment 1 (testing one condition takes 4 times longer in Experiment 1 than in Experiment 2). 

Discussion

L 443-444: “an experimental design usually leading to risk-aversion [26]”. But Ref 26 found risk aversion in bonbos and risk proneness in chimps and thus doesn’t fit the statement very well

Modification of the manuscript : lines 522-523.

L 452-454: “Finally, [28] could not reject that subjects understood the task and properly inferred the probabilities to win without experience. In this study, we fixed these experimental issues” -> explanation required how exactly the current study fixed the issue.

We modified the experimental design (counterbalancing the sides of safe and risky options, removing the refreshers trials, adding of a condition where the risky option is as large as to the safe option, using a low-level reward,...) which resulted in lower levels of risky choices, as expected, and allows to address [28] concern that subject did not understand the task, as we showed that they responded rationality to variations in probability or potential gain. This is mentioned in detail in the introduction of the manuscript, lines 144 to 155, and 176 to 180.

L 456-457: On the videos provided, it looks like the baiting procedure is well in sight of Experimenter 2. Was this always the case?

No, this was not the case: E2 stayed close to E1 but was not looking at where the food was hidden. 

L 457: “levels of risky choices” -> does this refer to the size of the risky option?

No, it refers to the percentage of risky choices (this expression was used before in the result section, but can be modified if not clear).

L 458-459: “probability to win to verify that subjects based their strategy on the value of that probability and were thus able to compute it” -> its not fully clear to which of the listed results this is referring.

Modification of the manuscript : lines 537 to 544.

L 512. What pilot study? It is mentioned the first time here, should the reader know about it already at this point?

The wording “pilot study” was incorrect, we are referring to the analysis of response times in both experiments: we measured subjects' response times for every trial of Experiment 1 and 2 to test whether there was a relationship between RT and experimental design. As the result was inconclusive (longer response times in Experiment 2, which could be interpreted as due to the fact that subjects had more cups to choose from).

The “lateral hypothesis”. The position bias is interesting, and also that the two species apparently had different kind of bias. However, what does this tell us about the risky choice results? Are they meaningful at all when subjects had a clear position bias? I think this point deserves mention and being discussed.

We addressed this point in the discussion (lines 244 to 247, and 255 to 258) : even though subjects had a clear position bias, this doesn't affect risk-assessment results in gorillas (as the side of the safe option was counterbalanced between trials). For orang-utans, the lateral bias apparently interacts with subjects' choice between the safe and risky option, but as the total number of risky cups is a good predictor of levels of risky choice, this lateral bias is not the only predictors of subjects' choices, as they are making rational economic decisions.

Reviewer #2: 

Uncertainty vs. risk tasks

My main comment concerns the difference between uncertainty/ambiguity and risk scenarios. The authors interpret their two studies as tasks suitable to test risk preferences. However, in their discussion, they comment that during the single-cup experiment 1, apes learn the probabilities by experience (as opposed to experiment 2, in which the exact probabilities are described since apes can observe the risky reward and the number of cups). While I agree with the distinction, I am surprised that the authors do not discuss differences in terms of ambiguity/risk. One possibility is that in experiment 1, it is harder to learn the probability of appearance despite the experience. Although the study reports that a rise in the probability to win had an increasing effect on apes risk choices, one could imagine that apes were often deciding under conditions of ambiguity (especially at the beginning of experiment 1 session), and that could partially explain differences between experiments 1 and 2 overall preferences towards risk. If that were the case, the results would also align with previous studies by Rosati & Hare, 2011 and Haux et al., 2021.

This figure shows the level of risky choices in E1 for the 4 sessions of each PxV combination. We cannot explain the difference between E1 and E2 by the sole hypothesis that subjects make their choices under ambiguity in E1, because, as subjects learn the probability (and the quantity of reward) throughout the 4 sessions of 10 trials for each PxV condition of E1, the level of risky choices does not drastically increase (on the contrary, the level of risky choices tends to decrease). 

L 99 and 105: I am not sure if the ref. 26 interpretation is accurate. The chimpanzees were risk-prone in that study. Furthermore, risk options always provided rewards (either less or more grapes than the secure option).

Modification of the manuscript : lines 95 to 119.

L 132: I would stick with rewards instead of awards.

Modification of the manuscript : line 146.

L 180: Check if the hypothesis are correct. The authors mention that they expect risk proneness in both tasks, but in L 99, they argue that primates are mostly risk-averse. I guess the authors expect more risk aversion or neutrality in experiment 1, as they have found.

Modification of the manuscript : lines 186, 187.

Training:

In general, some details are missing. Which were the quantities involved in the training? Did they need to reach a rate of 80% correct over how many trials?

Also, what was the purpose of the hidden condition, to demonstrate that they can remember the quantities through the transparent saucer before they are covered? Did they then also need to reach at least an 80% of success in those trials? Please specify this in the manuscript.

Modification of the manuscript : lines 234 to 247.

Experiment 1:

L 260: I wonder if the authors could analyze the session effect in experiment 1 as they did in experiment 2 (instead of having session as a random effect). A session effect could tell us whether apes were learning the EV across time—increasing or decreasing their risk choices depending on the probability of obtaining the reward (P) and its value (V). The last comment relates to my previous one on the difference between ambiguity/risk choices. 

Modification of the manuscript : lines 388 to 393.

It might also help to specify whether the presentation order varied between the four sessions. For instance, if the probability of food present in the risk choice was 0.5, was food present in the risk option every second trial, or was the presentation randomized as long as there would be a total of 5 trials with and without food? If trial presentation order was blocked within sessions, then ambiguity was reduced since the four sessions would be exactly the same.

For P=0.3, we assigned 1 (baited cup) in every 3 trials, ex : 0 1 0 0 0 1 0 1 0 1 0 0 0 1 0 1 0 0 1 (0 0). The last two trials were not performed as we conducted 20 trials per day. 

For P=0.25, we assigned 1 (baited cup) in every 4 trials, ex : 0 1 0 0 0 0 1 0 etc. 

For P=5, we assigned 1 (baited cup) in every 2 trials, ex : 0 1 0 1 1 0 0 11 0 0 1 etc. 

Ambiguity was then reduced as we limited the variability of the occurrence of baiting and baited the risky cup with a frequency that was highly similar to the probability. 

Statistical analysis:

The description of the models is very clear except for the fixed-effects part. The authors describe the random effects and the additional fixed effects for experiment 2, but it is unclear which are the shared fixed effects between studies (e.g., 318). It is only apparent in the result section (e.g., the value of the risky option).

Modification of the manuscript : lines 326 to 369.

Results:

I would remind the readers that the probability of the risky option refers to P in the table, that the value of the risky option refers to V and that the expected value refers to EV =P*V.

Modification of the manuscript : lines 382 to 384, and 416 to 418.

Discussion:

L 444: Ref 26 leads to risk aversion only in bonobos.

Modification of the manuscript : lines 522, 523.

L 487: Close parenthesis after reference 43.

Modification of the manuscript : line 572.

L 492: either "described" or 'described'.

Modification of the manuscript : line 577.

L 493 to 495: I find these sentences slightly unclear (e.g., "with feedback the chosen option").

Modification of the manuscript : line 579.

L 512: It might be interesting to report some descriptives of the pilot regarding the latencies discussed.

We analysed the response times in both experiments and received the following results (see figure below). 

L 563: I understand the logic in the possible distinction between the risk assessment task in the domain of gains (experiment 1) and the domain of losses (experiment 2). However, how do we know whether apes took the risk choice as the reference in experiment 2? Perhaps the authors could cite some prospect theory literature showing whether better options are usually taken as reference points?

We don't know whether the apes took the risk choice as the reference, but if they did, that would explain our result. We just brought up that hypothesis, but it would indeed have to be tested.

Also, the authors mention that, given that the risk option is the reference, choosing it leads to the loss of all the reference quantity—instead of a proportion of it when choosing the safe option. However, that might not be true for all trial constellations since sometimes the risk option is equally probable, and they can also obtain it from time to time. In other words, they do not permanently lose all the “reference” quantity. Otherwise, they would always take the safe option.

Indeed, the sentence was incorrect.

L 610: I would suggest making the potential role of single cup design more explicit.

Modification of the manuscript : line 693.

---

## [Decision Letter · Decision Letter 1]

21 Jan 2022

PONE-D-21-29220R1Rationality and cognitive bias in captive gorillas' and orang-utans' economic decision-makingPLOS ONE

Dear Dr. Lacombe,

Thank you for submitting your manuscript to PLOS ONE. After careful consideration, we feel that it has merit but does not fully meet PLOS ONE’s publication criteria as it currently stands. Therefore, we invite you to submit a revised version of the manuscript that addresses all the points raised by both reviewers during the review process.

We look forward to receiving your revised manuscript.

Kind regards,

Elsa Addessi

Academic Editor

PLOS ONE

Reviewers' comments:

Reviewer's Responses to Questions

**Comments to the Author**

1. If the authors have adequately addressed your comments raised in a previous round of review and you feel that this manuscript is now acceptable for publication, you may indicate that here to bypass the “Comments to the Author” section, enter your conflict of interest statement in the “Confidential to Editor” section, and submit your "Accept" recommendation.

Reviewer #1: (No Response)

Reviewer #2: (No Response)

2. Is the manuscript technically sound, and do the data support the conclusions?

Reviewer #1: (No Response)

Reviewer #2: Yes

3. Has the statistical analysis been performed appropriately and rigorously? 

Reviewer #1: (No Response)

Reviewer #2: Yes

4. Have the authors made all data underlying the findings in their manuscript fully available?

Reviewer #1: (No Response)

Reviewer #2: Yes

5. Is the manuscript presented in an intelligible fashion and written in standard English?

Reviewer #1: (No Response)

Reviewer #2: Yes

6. Review Comments to the Author

Reviewer #1: First of all, I would like to thank the authors for addressing my previous comments thoroughly. While I am going to list a number of points that I still think should be addressed, I want to emphasize that I find the manuscript is already much improved and the authors’ effort to accommodate Editor’s and Reviewers’ previous comments is clearly visible.

In brief, my main points concern the analysis. Particularly missing random slopes and additional t-tests. If the results hold when random slopes are included in the models, or if the authors can justify why they do not need a random slopes structure in their analysis (maybe I overlooked something in the design of the study?), then that’s great and I have no other major concerns.

In the following, I list my points of concern and questions in more detail.

Statistical models

• L 339-340: “The significance of each predictor variable in explaining variation in rate of risky choices was tested with ANOVA” -> Could the authors clarify what they mean with this in the context of a full-null model comparison approach? In my understanding, when model comparison found that two models differ then the model output statistics tell about contribution of each predictor to explain variation and no additional ANOVAs are needed.

• L 335-343: to which analysis is this model referring? If this is meant as a general paragraph to report that assumption checks etc were done for each of the analyses, please clarify accordingly. The paragraph about full-null model comparisons should be moved up here.

• L 345-354: description of the full model to analyze Exp.1. Just minor, but this was a bit difficult to understand, because on the one hand the authors count predictor variables (up to 5), but on the other hand they have more predictor terms in their full model, namely the additional two 2-way interactions. Maybe this could be phrased a bit differently, something like this: Our full model included x and y … as well as their interaction as fixed predictor terms of interest. To control for z and xx…, we also included these as fixed effects but kept them in the null model.

• L 373: Shouldn’t species be a predictor of interest, why was it kept in the null model? I understood a potential species difference was one of the main research questions.

• L 357-361: Is this part of the original confirmatory analysis plan to test the research question of risk preference? It sounds more like a position/side bias check. It would be helpful if this was clarified explicitly.

o This reminds me of a methodological question: according to which logic were hidden reward positions determined? I assume this was somehow balanced such that each of the four possible risky positions held a risky cup equally often and a reward was hidden at each position equally often across the different number of risky cup trials? Or were risky cups always positioned immediately next to the safe cup with no empty positions when fewer than 4 risky cups were presented? It information would be useful for readers who want to use the paradigm and replicate the study.

• L 376-381: The point of modelling risk preference as a function of the above specified predictor variables is to account for the influence of these predictor variables. The described t-tests ignore these aspects. Assuming, for example, the t-test reveals risk neutrality, but the GLMM shows a clear effect of session (for example, apes are risk averse at the beginning and become increasingly risk-seeking throughout the study) – Then what have you gained with this t-test? It provides less and potentially misleading information. Therefore, I find the described t-tests inappropriate here.

• I am still missing random slopes in the model structure. Random effects need to be considered when individuals provide repeated observations. But here, individuals contribute repeated observations under different repeated conditions (i.e., the different values & probabilities of risky option, different sessions in which these conditions are tested for every individual, etc), which should be modeled via appropriate random slopes. In this case, I think one would want to include random slopes of all predictor terms of interest as well as session within individual and of all predictor terms of interest within session. (See e.g. Barr et al, 2013, for why random slopes are important to draw reliable inference about fixed effects 10.1016/j.jml.2012.11.001)

Results

• In general, I would like to see the models run with random slope structure to see if the results hold; nonetheless, I have some comments on the current results.

• L 390 – 398: The authors report about main effects of value of risky option, probability to win and session but then report the two interactions including these variables turned out significant. This means the main effects should not be interpreted because of limited use when they interact with other variables.

• L 400-404: I suppose these results are based on the additional t-tests not on the model comparison, because the authors report earlier that species remained a predictor in the null model. It would help if it was mentioned again to the reader to which analysis the respective results belong. (but see my general critique of the t-tests above)

• L 481-486: If risky choice was affected by an interaction of type of experiment and value of the risky option, then the main effect of experimental design shouldn’t be interpreted (l 481-482).

• L 485-487 this sentence doesn’t seem to reflect the model results and Fig. 4B. Model results show no interaction between experimental design and probability to win, and Fig. 4B depicts something that looks more like a main effect of experiment (with more risky choice in Exp. 2) with the exception of when p =1, to which individuals seem to be sensitive in Exp. 2 but not Exp. 1.

Discussion

• L 524-526: “in particular, if controlled for expected gains, subjects were generally risk-neutral in Experiment 1 and risk-prone in Experiment 2.” I don’t know what the authors mean by this and how they controlled for expected gains. Is this referring to section “Performance in 'single cup' vs 'multiple cup' design”? If yes, please clarify whether main effect of experimental design can be interpreted despite interacting with risk probability. Also, why is expected gain a non-economic parameter?

• L 543: For clarification that this paragraph refers specifically to Exp. 2, add “In this study, we addressed these experimental issues of the multiple cup design”

• 551: remove “did”

• 628-630: “However we found no support due to the fact that both species exhibited positional biases (gorillas: lateral bias; orang-utans: central bias) in Experiment 2”. For clarification, add a sentence explaining what you mean. For example, something like “Specifically, rather than exploring the full range of possible locations, gorillas exhibited a strong preference for the left-most location and orangutans for the central locations.”

Figure S3: there is no figure being displayed for me in the supplementary document, there is a white picture but nothing else

The authors have provided figures of individual performance in their response letter, but why not include them in the ESM for interested readers? For the figure of individual response Exp. 2: are error bars displayed correctly? It seems some values do have error bars while others don’t have them – or are they so small that they don’t display well? Its especially apparent in Figure C.

Reviewer #2: Overall, the manuscript is clearer now. However, I still find inconsistencies in how the statistical analysis are described and the results reported. I also have a few other comments in other manuscript sections that should be addressed before the final acceptance.

Introduction

L 135. The authors stated in ref. 39 that bonobos chose the risky option in 87.5% of trials. The numbers do not seem to coincide with figure 2 of ref. 39 unless the authors are averaging between small and medium sizes. Is this the case? Please clarify.

Methods

L 274. Table 1 seems to miss a minus value "(-)" after EV 1.

Statistical analysis

L 341: "we reached a model with interpretable terms".

L 356: I believe the added predictor is a sixth rather than a third predictor in the model. Experiment 2 analysis contains the same 5 predictors of Experiment 1 (risk probability, risk value, specie, session and side of the safe cup) + the addition of type of reward.

L 360: There seems to be an interaction between the position of the cup and species, as indicated in the results and in Table 5. This information is missing here, together with the fact that species was also a predictor.

L 372: I am confused because the null models included the species variable. To my understanding, that means the authors did not analyze the effect of species statistically (a full model including species against a null model without the variable) in any of their models, and therefore species differences should not be reported.

Results

General comment: The results are more precise than in the previous version of the manuscript, but I still have considerations about how they are reported. The authors constantly report main effects even when interactions between those main effects are significant. This is especially salient in Experiment 1. Probability to win, value of the risky option and session are all significant. Similarly, the interactions between session * probability and session * value are also significant. The authors should only report the interaction' effects and plot the results accordingly. In other words, the predictors' effect only makes sense in interaction with each other, not on their own anymore.

L 397: Please report more information on the significant two-way interaction. In which direction did they modify their choices?

Table 3: The inclusion of tables is helpful, but I would suggest the authors report more information regarding degrees of freedom, confidence intervals (values that are already calculated since they are plotted), and the model's estimates.

L 431: "..we found that the the session"

L 433: I would inform readers about the interaction between value and session. Also, I would suggest first reporting the effect of session in L 432 (that subjects increase their risk tendency over the experiment sessions) and then the non-significant effect of the interaction, clarifying that the apes did not learn the probabilities to win over repeated testing.

L 465: Same general comment I made earlier. If an interaction between the position of the cup and species is significant, the position of the cup on the trolley per se adds no value to the interpretation of the results.

L 483: The significant interaction between the type of experiment and the value of the risky option is not reported in table 6. In addition, what do the authors exactly mean by "subjects were more attentive to the value of the risky option in exp 1 than 2". It is unclear that attentive means that apes chose the risky option less in exp 1 compared to exp 2, as shown in figure 4A.

L 488; To clarify, trials with P = 1 refer to any trial with just two cups regardless of the number of rewards in the risky cup, right?

Discussion

L 525: I would add that controlled for expected gains means when EV = 2.

L 592: References missing for those experiments from experience providing partial feedback.

L 606: I would include the analysis in the manuscript. The authors reported the resulting plots to both reviewers in the previous review round. I think they should report the results of the analysis and the plots, at least in the supplementary materials, with a reference in the results section.

L 686: I would say that there were no apparent differences between species since the authors previously report an almost significant effect of species (p = 0.052) in Experiment 1 (although the authors may not require reporting species differences if their effect was controlled for and thus not statistically analyzed).

7. PLOS authors have the option to publish the peer review history of their article (what does this mean?). If published, this will include your full peer review and any attached files.

Reviewer #1: No

Reviewer #2: No

---

## [Author Response · Author response to Decision Letter 1]

7 Apr 2022

Reviewer #1:

In brief, my main points concern the analysis. Particularly missing random slopes and additional t-tests. If the results hold when random slopes are included in the models, or if the authors can justify why they do not need a random slopes structure in their analysis (maybe I overlooked something in the design of the study?), then that’s great and I have no other major concerns.

Random slopes were added to our model and we compared the full model with random slopes to simpler model removing random slopes one by one in order to keep the most parsimonious one. See lines 339->344 and supplementary tables S2, S6 and S9.

L 339-340: “The significance of each predictor variable in explaining variation in rate of risky choices was tested with ANOVA” -> Could the authors clarify what they mean with this in the context of a full-null model comparison approach? In my understanding, when model comparison found that two models differ then the model output statistics tell about contribution of each predictor to explain variation and no additional ANOVAs are needed.

The phrasing we used was confusing. We did not performed an “ANOVA” but an analysis of deviance (type II Wald chisquare test) in order to test the significance of predictors. We removed the comparison to the null model from our analysis.

• L 335-343: to which analysis is this model referring? If this is meant as a general paragraph to report that assumption checks etc were done for each of the analyses, please clarify accordingly. The paragraph about full-null model comparisons should be moved up here.

Is is indeed a general paragraph We made it more clear (line 324).

L 345-354: description of the full model to analyze Exp.1. Just minor, but this was a bit difficult to understand, because on the one hand the authors count predictor variables (up to 5), but on the other hand they have more predictor terms in their full model, namely the additional two 2-way interactions. Maybe this could be phrased a bit differently, something like this: Our full model included x and y … as well as their interaction as fixed predictor terms of interest. To control for z and xx…, we also included these as fixed effects but kept them in the null model.

We made it more clear (lines 334 to 339)

L 373: Shouldn’t species be a predictor of interest, why was it kept in the null model? I understood a potential species difference was one of the main research questions.

Indeed species should not be kept in the null model. In any case, the comparison to the null model was removed (we tested the significance of predictors with analysis of deviance).

L 357-361: Is this part of the original confirmatory analysis plan to test the research question of risk preference? It sounds more like a position/side bias check. It would be helpful if this was clarified explicitly.

We made it more clear (line 362).

This reminds me of a methodological question: according to which logic were hidden reward positions determined? I assume this was somehow balanced such that each of the four possible risky positions held a risky cup equally often and a reward was hidden at each position equally often across the different number of risky cup trials? Or were risky cups always positioned immediately next to the safe cup with no empty positions when fewer than 4 risky cups were presented? It information would be useful for readers who want to use the paradigm and replicate the study.

Indeed, we balanced the position of the baited risky cup so that each risky cup help the reward equally often. We made it more clear (lines 290->291).

L 376-381: The point of modelling risk preference as a function of the above specified predictor variables is to account for the influence of these predictor variables. The described t-tests ignore these aspects. Assuming, for example, the t-test reveals risk neutrality, but the GLMM shows a clear effect of session (for example, apes are risk averse at the beginning and become increasingly risk-seeking throughout the study) – Then what have you gained with this t-test? It provides less and potentially misleading information. Therefore, I find the described t-tests inappropriate here.

Indeed, we removed that analysis from our manuscript. We tested risk-preference at the indifference point using post-hoc tests on our models (see lines 345->352, and result section).

I am still missing random slopes in the model structure. Random effects need to be considered when individuals provide repeated observations. But here, individuals contribute repeated observations under different repeated conditions (i.e., the different values & probabilities of risky option, different sessions in which these conditions are tested for every individual, etc), which should be modeled via appropriate random slopes. In this case, I think one would want to include random slopes of all predictor terms of interest as well as session within individual and of all predictor terms of interest within session. (See e.g. Barr et al, 2013, for why random slopes are important to draw reliable inference about fixed effects 10.1016/j.jml.2012.11.001)

Random slopes were added to our model and we compared the full model with random slopes to simpler model removing random slopes one by one in order to keep the most parsimonious one. See lines 339->344 and supplementary tables S2, S6 and S9.

In general, I would like to see the models run with random slope structure to see if the results hold; nonetheless, I have some comments on the current results.

Random slopes were added to our model and we compared the full model with random slopes to simpler model removing random slopes one by one in order to keep the most parsimonious one. See lines 339->344 and supplementary tables S2, S6 and S9.

L 390 – 398: The authors report about main effects of value of risky option, probability to win and session but then report the two interactions including these variables turned out significant. This means the main effects should not be interpreted because of limited use when they interact with other variables.

Indeed, we modified the manuscript and report the main effects but do not interpret it anymore.

L 400-404: I suppose these results are based on the additional t-tests not on the model comparison, because the authors report earlier that species remained a predictor in the null model. It would help if it was mentioned again to the reader to which analysis the respective results belong. (but see my general critique of the t-tests above)

We modified that analysis and do not perform t-test anymore (see lines 406-413).

L 481-486: If risky choice was affected by an interaction of type of experiment and value of the risky option, then the main effect of experimental design shouldn’t be interpreted (l 481-482).

We modified the manuscript and removed that interpretation.

L 485-487 this sentence doesn’t seem to reflect the model results and Fig. 4B. Model results show no interaction between experimental design and probability to win, and Fig. 4B depicts something that looks more like a main effect of experiment (with more risky choice in Exp. 2) with the exception of when p =1, to which individuals seem to be sensitive in Exp. 2 but not Exp. 1.

Indeed there was a mistake in the sentence, we modified the manuscript (line 471->474).

L 524-526: “in particular, if controlled for expected gains, subjects were generally risk-neutral in Experiment 1 and risk-prone in Experiment 2.” I don’t know what the authors mean by this and how they controlled for expected gains. 

We made it more clear (see line 515).

Is this referring to section “Performance in 'single cup' vs 'multiple cup' design”? If yes, please clarify whether main effect of experimental design can be interpreted despite interacting with risk probability. Also, why is expected gain a non-economic parameter?

Yes it is referring to that section. In our opinion, the effect of experimental design is multiple : 

1) there is an overall higher level of risky choices in Experiment 2 than in Experiment 1 (see Fig. 4 and post-hoc tests)

2) the experimental design affects subjects' perception of the P=1 condition (see Fig. 4)

3) the experimental design affects subjects' perception of the value of the risky option (analysis of deviance in table 6 and Fig. 4)

We did not mean than expected gain is a non-economic parameter. The wording was not clear, we were only referring to experimental design when we mentioned “non-economic parameters”.

L 543: For clarification that this paragraph refers specifically to Exp. 2, add “In this study, we addressed these experimental issues of the multiple cup design”

We made it more clear (line 533)

551: remove “did”

Done

628-630: “However we found no support due to the fact that both species exhibited positional biases (gorillas: lateral bias; orang-utans: central bias) in Experiment 2”. For clarification, add a sentence explaining what you mean. For example, something like “Specifically, rather than exploring the full range of possible locations, gorillas exhibited a strong preference for the left-most location and orangutans for the central locations.”

We made it more clear (lines 618-620).

Figure S3: there is no figure being displayed for me in the supplementary document, there is a white picture but nothing else

Fixed 

The authors have provided figures of individual performance in their response letter, but why not include them in the ESM for interested readers? For the figure of individual response Exp. 2: are error bars displayed correctly? It seems some values do have error bars while others don’t have them – or are they so small that they don’t display well? Its especially apparent in Figure C.

Indeed we fixed the error bars. The figures are now in the SM.

Reviewer #2: 

L 135. The authors stated in ref. 39 that bonobos chose the risky option in 87.5% of trials. The numbers do not seem to coincide with figure 2 of ref. 39 unless the authors are averaging between small and medium sizes. Is this the case? Please clarify.

We selected the trials in ref. 39 where the safe and the risky option had the same expected value (in this article, the indifference point is at EV=1). According to table 1 of ref. 39, this only corresponds to trials where the safe option was medium (V=3) and their was 3 risky cups (P=0.33). Going back to table 2 this corresponds to a level of risky choice of 75% (indeed, not 87,5%) for bonobos and 100% for chimpanzees.

L 274. Table 1 seems to miss a minus value "(-)" after EV 1.

Done

L 341: "we reached a model with interpretable terms".

Done

L 356: I believe the added predictor is a sixth rather than a third predictor in the model. Experiment 2 analysis contains the same 5 predictors of Experiment 1 (risk probability, risk value, specie, session and side of the safe cup) + the addition of type of reward.

We modified it (line 354).

L 360: There seems to be an interaction between the position of the cup and species, as indicated in the results and in Table 5. This information is missing here, together with the fact that species was also a predictor.

We modified it.

L 372: I am confused because the null models included the species variable. To my understanding, that means the authors did not analyze the effect of species statistically (a full model including species against a null model without the variable) in any of their models, and therefore species differences should not be reported.

Indeed species should not be kept in the null model. In any case, the comparison to the null model was removed (we tested the significance of predictors with analysis of deviance).

General comment: The results are more precise than in the previous version of the manuscript, but I still have considerations about how they are reported. The authors constantly report main effects even when interactions between those main effects are significant. This is especially salient in Experiment 1. Probability to win, value of the risky option and session are all significant. Similarly, the interactions between session * probability and session * value are also significant. The authors should only report the interaction' effects and plot the results accordingly. In other words, the predictors' effect only makes sense in interaction with each other, not on their own anymore.

Indeed, we modified the manuscript following your suggestion (see lines 395->405).

L 397: Please report more information on the significant two-way interaction. In which direction did they modify their choices?

See lines 395->405 and tables S4 and S5.

Table 3: The inclusion of tables is helpful, but I would suggest the authors report more information regarding degrees of freedom, confidence intervals (values that are already calculated since they are plotted), and the model's estimates.

See tables 3, 4, 5, 6 and tables S3, S8.

L 431: "..we found that the the session"

Done

L 433: I would inform readers about the interaction between value and session. Also, I would suggest first reporting the effect of session in L 432 (that subjects increase their risk tendency over the experiment sessions) and then the non-significant effect of the interaction, clarifying that the apes did not learn the probabilities to win over repeated testing.

Modified following your suggestion (434->441).

L 465: Same general comment I made earlier. If an interaction between the position of the cup and species is significant, the position of the cup on the trolley per se adds no value to the interpretation of the results.

We modified it.

L 483: The significant interaction between the type of experiment and the value of the risky option is not reported in table 6. In addition, what do the authors exactly mean by "subjects were more attentive to the value of the risky option in exp 1 than 2". It is unclear that attentive means that apes chose the risky option less in exp 1 compared to exp 2, as shown in figure 4A

We were referring to the slope of the curve Choice = f(value risky option), that is steeper in Experiment 1 than in Experiment 2, suggesting that subjects were more attentive to variations of the value of risky option in Experiment 1. However, we removed that interpretation.

L 488; To clarify, trials with P = 1 refer to any trial with just two cups regardless of the number of rewards in the risky cup, right?

Yes, any trial with only the safe and one risky cup.

L 525: I would add that controlled for expected gains means when EV = 2.

Done

L 592: References missing for those experiments from experience providing partial feedback.

Done (line 581).

L 606: I would include the analysis in the manuscript. The authors reported the resulting plots to both reviewers in the previous review round. I think they should report the results of the analysis and the plots, at least in the supplementary materials, with a reference in the results section.

Done (lines 487 to 493 and figure S9).

L 686: I would say that there were no apparent differences between species since the authors previously report an almost significant effect of species (p = 0.052) in Experiment 1 (although the authors may not require reporting species differences if their effect was controlled for and thus not statistically analyzed).

Done (lines 673 to 675).

---

## [Decision Letter · Decision Letter 2]

19 May 2022

PONE-D-21-29220R2Rationality and cognitive bias in captive gorillas' and orang-utans' economic decision-makingPLOS ONE

Dear Dr. Lacombe,

Thank you for submitting your manuscript to PLOS ONE. After careful consideration, we feel that it has merit but does not fully meet PLOS ONE’s publication criteria as it currently stands. Therefore, we invite you to submit a revised version of the manuscript that addresses the points raised during the review process.

 Please very carefully take into account the additional remarks on statistical analyses provided by both reviewers, along with some minor suggestions, reported below. 

We look forward to receiving your revised manuscript.

Kind regards,

Elsa Addessi

Academic Editor

PLOS ONE

Journal Requirements:

Additional Editor Comments:

- abstract: Pongo abelii should be in italics

- l 275: “Probability of obtaining the reward when choosing the risky option” rather than “Probability of winning the reward in the risky option”

ll 278-9: “each condition (box) consisted of 40 trials”

l 315: “each condition (box) consisted of 10 non-consecutive trials”

l 320: “we did not perform consecutively” (rather than “in a row”)

l 321: “was tested in two consecutive sessions”

l 348: why was session number fitted as a categorical variable rather than a continuous variable?

l 376: “where the response variable was”

l 397: “showed that the main effects of the probability and the value of the risky options were significant (…) as well as those of the species (…) (please make a similar change also on ll 445-7 and 485-7)

ll 399 & 402: Actually, p = 0.051 or 0.052 are marginally significant, please rephrase on ll 399 and 402

l 413: please erase “that” before “the trend estimates”

l 447: “When investigating the effect of session on subjects’ choices”

l 456: “there was no difference” (please erase “specific”)

l 472: “left location” (please erase “most”)

l 486: please replace “the value to win” with “the value of the risky option”, here and wherever else applicable

l 497: “To investigate”

l 504: please erase “the” (typo)

ll 533-6: please add here a caveat concerning the small sample size of the present study

l 539 Actually, capuchin monkeys were risk prone in this task (please check “F De Petrillo, M Ventricelli, G Ponsi, E Addessi 2015 Do tufted capuchin monkeys play the odds? Flexible risk preferences in Sapajus spp. Animal Cognition 18 (1), 119-130)

ll 577-8: please rephrase as follows: “decision-makers could experience probability distortions depending on how probabilities are presented to them”

l 581: “experiment based on experience”…”experiment based on description”

l 690: please erase “their” before “preliminary conclusions”

Reviewers' comments:

Reviewer's Responses to Questions

**Comments to the Author**

1. If the authors have adequately addressed your comments raised in a previous round of review and you feel that this manuscript is now acceptable for publication, you may indicate that here to bypass the “Comments to the Author” section, enter your conflict of interest statement in the “Confidential to Editor” section, and submit your "Accept" recommendation.

Reviewer #1: (No Response)

Reviewer #2: All comments have been addressed

2. Is the manuscript technically sound, and do the data support the conclusions?

Reviewer #1: Partly

Reviewer #2: Yes

3. Has the statistical analysis been performed appropriately and rigorously? 

Reviewer #1: No

Reviewer #2: Yes

4. Have the authors made all data underlying the findings in their manuscript fully available?

Reviewer #1: Yes

Reviewer #2: Yes

5. Is the manuscript presented in an intelligible fashion and written in standard English?

Reviewer #1: Yes

Reviewer #2: Yes

6. Review Comments to the Author

Reviewer #1: I have given general and specific comments to all parts of the manuscript in previous rounds of the review process and will focus on the statistics in this review.

I can see the authors worked on taking on board my and the other Reviewer’s previous comments, but still have concerns about the reported analysis.

I still can’t see a global model comparison to evaluate the contribution of the fixed predictor terms. The authors report they performed an analysis of deviance and did a model reduction by excluding random terms and nonsignificant fixed effects interaction terms. However, this process is not documented sufficiently. For example, in Table 3 we see that the three-way interactions are „significant“, but apparently some lower-level two-way interactions have been removed. It is unclear for me how the final model looked like. When higher-level interactions are included in a model, lower-level interaction terms are automatically included as well. In l 340 it says „We removed non-significant interaction terms one by one until we reached a model with interpretable terms (without non-significant interaction terms)“. But this process is not sufficiently documented (which terms were removed; at what point/which order; based on comparison with which model/criterion?).

The authors compare models with different random slopes structures, but I can’t see for what purpose this is done. Generally, a maximum random slopes structure is recommended and for comparison of models to evaluate the fixed predictor terms, the compared models should have the same random slopes structure (Barr et al., 2013. https://doi.org/10.1016/j.jml.2012.11.001).

Hence, what I would like to see is a comparison of the full model (fixed effects of interest [+ optional fixed effects of variables to only control for] + full random effect/slope structure) with a null model ([optional fixed effects to only control for] + full random effect/slope structure). If this comparison indicates that the full model doesn’t explain the data better than the null model, then that is the result. If this comparison indicates that the full model explains the data better than the null model, then the effects of the predictor terms can be assessed and potentially one continues to reduce fixed effects structure by removing nonsignificant interactions, starting with the highest-level interaction. But this global full-null model comparison is not reported, so we don’t even know if those significant terms even matter to explain the data better. See here, for example, for why model full model comparisons are important (Forstmeier & Schielzeth, 2011. 10.1007/s00265-010-1038-5 10.1007/s00265-010-1038-5).

Why is effect of species tested as part of an interaction in Exp. 1 but not in Exp. 2?

L 355: „If needed we ran post-hoc tests to calculate estimated marginal means or estimated trends“. Could you specify what „if needed“ means, please.

In general, I am a bit concerned that the authors might want too much of their data. Considering that only five resp. three individuals per species were tested, including two three-way-interactions and corresponding random slopes structure constitutes in a massively complex model! I would be interested in information about effect sizes and model stability (i.e., comparing the estimates from the model based on all data with those from models with the levels of the random effects excluded one at a time). Should it turn out that not all questions can be answered at once with the existing data set, then it would be better to simplify the analysis and reduce the number of research questions.

L 388-391 „The final model without random slopes, and with individual and session within individual as random intercepts (see supplementary table S2) was not significantly different from the full model (LRT: χ2(4)=3.83, p=.43)“. This sounds circular to me. It’s intrinsic to the process of removing terms previously and seems kind of circular. When I have a model and remove stuff that is „not significant“ and then compare this reduced model to the previous model, it’s expected that the difference is not significant again (because this was the reason why the term was removed in the first place).

L 340-341: Non-significant interaction terms should not be called „uninterpretable“.

Supplementary Figures. Thank you for including Figures on individual data. Figures S6 and S7 are very busy. Would it be possible to split the data of the two species into two parallel plots respectively? In addition, it would help to have two different symbols representing the species level in addition to different line/symbol colours, wherever both species are presented in the same plot. And it would help if a colour legend would be displayed along with every Figure. The information given in Table S1 but it‘s very unintuitive having to look this up some pages down in a table.

Fig. S4+S5: I would find it more intuitive to present session on the x-axis and display the value/probability to win of risky option as different lines.

I have noted a few spelling and grammatical errors throughout, which I personally wouldn’t mind (not being a native English speaker myself); but since Plos One specifically advices to comment on this, I recommend to run a spell and grammar check across the paper bevor a final submission.

Reviewer #2: The paper has improved and I am willing to accept the manuscript, after few minor comments and one clarification is addressed.

Main clarification point: I applaud the inclusion of the random slopes, the clarity in the interpretation of the results, and in general all statistical changes the authors have addressed. The methods and results sections are much clearer in the current version. However, although I do not think the following information is necessary to be included in an accepted version of the manuscript—I leave this decision to the editor, I would like to know why the authors did no longer use the full-null model comparison LRT method (as they do with full and final models) and instead tested the significance of their predictors with an analysis of deviance. Such a drastic change is worth an explanation before acceptance, especially when functions such as drop1 in lme4 package allows to obtain the p-values for single and interaction predictors with the LRT framework.

Besides, in L 376 there seems to be a typo: “with the response variable was the proportion”.

Also, in conclusions L 744 “are currently unknown are require..”

7. PLOS authors have the option to publish the peer review history of their article (what does this mean?). If published, this will include your full peer review and any attached files.

Reviewer #1: No

Reviewer #2: No

---

## [Author Response · Author response to Decision Letter 2]

6 Jul 2022

Editor : 

- abstract: Pongo abelii should be in italics

- l 275: “Probability of obtaining the reward when choosing the risky option” rather than “Probability of winning the reward in the risky option”

ll 278-9: “each condition (box) consisted of 40 trials”

l 315: “each condition (box) consisted of 10 non-consecutive trials”

l 320: “we did not perform consecutively” (rather than “in a row”)

l 321: “was tested in two consecutive sessions”

Done

l 348: why was session number fitted as a categorical variable rather than a continuous variable?

Session was indeed fitted as a continuous variable, we corrected it.

l 376: “where the response variable was”

l 397: “showed that the main effects of the probability and the value of the risky options were significant (…) as well as those of the species (…) (please make a similar change also on ll 445-7 and 485-7)

ll 399 & 402: Actually, p = 0.051 or 0.052 are marginally significant, please rephrase on ll 399 and 402

l 413: please erase “that” before “the trend estimates”

l 447: “When investigating the effect of session on subjects’ choices”

l 456: “there was no difference” (please erase “specific”)

l 472: “left location” (please erase “most”)

l 486: please replace “the value to win” with “the value of the risky option”, here and wherever else applicable

l 497: “To investigate”

l 504: please erase “the” (typo)

ll 533-6: please add here a caveat concerning the small sample size of the present study

Done

l 539 Actually, capuchin monkeys were risk prone in this task (please check “F De Petrillo, M Ventricelli, G Ponsi, E Addessi 2015 Do tufted capuchin monkeys play the odds? Flexible risk preferences in Sapajus spp. Animal Cognition 18 (1), 119-130)

We specified it (line 107 and 531).

ll 577-8: please rephrase as follows: “decision-makers could experience probability distortions depending on how probabilities are presented to them”

l 581: “experiment based on experience”…”experiment based on description”

l 690: please erase “their” before “preliminary conclusions”

Done

Reviewer 1 :

I still can’t see a global model comparison to evaluate the contribution of the fixed predictor terms. 

We reverted to the LRT analysis in order to compare our full model to a null model (same random structure without fixed effects), then obtained a final model by removing non-significant interaction terms based on an analysis of deviance (type II Wald chisquare test).

The authors report they performed an analysis of deviance and did a model reduction by excluding random terms and nonsignificant fixed effects interaction terms. However, this process is not documented sufficiently. In l 340 it says „We removed non-significant interaction terms one by one until we reached a model with interpretable terms (without non-significant interaction terms)“. But this process is not sufficiently documented (which terms were removed; at what point/which order; based on comparison with which model/criterion?).

We made it more clear in the current version of the manuscript.

For example, in Table 3 we see that the three-way interactions are „significant“, but apparently some lower-level two-way interactions have been removed. It is unclear for me how the final model looked like. When higher-level interactions are included in a model, lower-level interaction terms are automatically included as well. 

We made it more clear. For every analysis (and for the null, full and final model), the random and fixed structures are presented in the supplemental material.

The authors compare models with different random slopes structures, but I can’t see for what purpose this is done. Generally, a maximum random slopes structure is recommended and for comparison of models to evaluate the fixed predictor terms, the compared models should have the same random slopes structure (Barr et al., 2013. https://doi.org/10.1016/j.jml.2012.11.001).

We did this in order to select the model with the more parsimonious random structure, but we removed that process in the current version and kept the complete random structure as you suggested.

Hence, what I would like to see is a comparison of the full model (fixed effects of interest [+ optional fixed effects of variables to only control for] + full random effect/slope structure) with a null model ([optional fixed effects to only control for] + full random effect/slope structure). If this comparison indicates that the full model doesn’t explain the data better than the null model, then that is the result. If this comparison indicates that the full model explains the data better than the null model, then the effects of the predictor terms can be assessed and potentially one continues to reduce fixed effects structure by removing nonsignificant interactions, starting with the highest-level interaction. But this global full-null model comparison is not reported, so we don’t even know if those significant terms even matter to explain the data better. See here, for example, for why model full model comparisons are important (Forstmeier & Schielzeth, 2011. 10.1007/s00265-010-1038-5 10.1007/s00265-010-1038-5).

We followed your suggestion and did this process for every analysis.

Why is effect of species tested as part of an interaction in Exp. 1 but not in Exp. 2?

The effect of species was tested as part of an interaction in both experiments but the effect was non-significant in Experiment 2 only, thus was removed from the final model. See supplemental tables S3 and S5 for the full models that we used in both experiments. 

L 355: „If needed we ran post-hoc tests to calculate estimated marginal means or estimated trends“. Could you specify what „if needed“ means, please.

We meant that if the main effect of a predictor (ex : session) was significant, we calculated EMM (for instance) for every level of that predictor. 

In general, I am a bit concerned that the authors might want too much of their data. Considering that only five resp. three individuals per species were tested, including two three-way-interactions and corresponding random slopes structure constitutes in a massively complex model! 

We added three-way interactions in order to answer questions from the reviewer in the previous rounds of reviews. The random slopes structure was also added at reviewers' suggestions. Would you wish that we simplify our models ?

I would be interested in information about effect sizes and model stability (i.e., comparing the estimates from the model based on all data with those from models with the levels of the random effects excluded one at a time). 

We followed your suggestion and selected the model with the more complete random structure.

L 388-391 „The final model without random slopes, and with individual and session within individual as random intercepts (see supplementary table S2) was not significantly different from the full model (LRT: χ2(4)=3.83, p=.43)“. This sounds circular to me. It’s intrinsic to the process of removing terms previously and seems kind of circular. When I have a model and remove stuff that is „not significant“ and then compare this reduced model to the previous model, it’s expected that the difference is not significant again (because this was the reason why the term was removed in the first place).

We followed your suggestion and selected the model with the more complete random structure.

L 340-341: Non-significant interaction terms should not be called „uninterpretable“.

Modified

Figures S6 and S7 are very busy. Would it be possible to split the data of the two species into two parallel plots respectively? In addition, it would help to have two different symbols representing the species level in addition to different line/symbol colours, wherever both species are presented in the same plot. And it would help if a colour legend would be displayed along with every Figure. The information given in Table S1 but it‘s very unintuitive having to look this up some pages down in a table.

Done

Fig. S4+S5: I would find it more intuitive to present session on the x-axis and display the value/probability to win of risky option as different lines.

Done

Reviewer #2

 I would like to know why the authors did no longer use the full-null model comparison LRT method (as they do with full and final models) and instead tested the significance of their predictors with an analysis of deviance. Such a drastic change is worth an explanation before acceptance, especially when functions such as drop1 in lme4 package allows to obtain the p-values for single and interaction predictors with the LRT framework.

We reverted to the LRT analysis in order to compare our full model to a null model (same random structure without fixed effects), then obtained a final model by removing non-significant interaction terms based on an analysis of deviance (type II Wald chisquare test).

Besides, in L 376 there seems to be a typo: “with the response variable was the proportion”.

Also, in conclusions L 744 “are currently unknown are require..”

Done

---

## [Decision Letter · Decision Letter 3]

26 Jul 2022

PONE-D-21-29220R3Rationality and cognitive bias in captive gorillas' and orang-utans' economic decision-makingPLOS ONE

Dear Dr. Lacombe,

Thank you for submitting your manuscript to PLOS ONE. After careful consideration, we feel that it has merit but does not fully meet PLOS ONE’s publication criteria as it currently stands. Therefore, we invite you to submit a revised version of the manuscript that addresses the points raised during the review process.

Please address very carefully Reviewer's 1 final remars on statistical analyses, namely provide an explicit assessment of model stability (and, as required, a caveat in the Discussion section in case your models shouldn't be stable), a revised analysis on log-transformed latencies and the required revisions on the Results section. Additionally, please proof-read the manuscript once again, as there are a few typos here and there. For instances, on l 373, "low-level reward" should be "low-valued reward".  Finally, I strongly advise to check out and cite the following papers, which are relevant for your literature review and relative hypotheses:

J Roig, A., Meunier, H., Poulingue, E., Marty, A., Thouvarecq, R., & Rivière (2022) Is economic risk proneness in young children (Homo sapiens) driven by exploratory behavior ? A comparison with capuchin monkeys (Sapajus apella). Journal of Comparative Psychology

J Rivière, A Kurt, H Meunier (2019) Choice under risk of gain in tufted capuchin monkeys (Sapajus apella): A comparison with young children (Homo sapiens) and mangabey monkeys (Cercocebus torquatus torquatus). Journal of Neuroscience, Psychology, and Economics 12 (3-4), 159                

J Rivière, M Stomp, E Augustin, A Lemasson, C Blois‐Heulin (2018) Decision‐making under risk of gain in young children and mangabey monkeys. Developmental Psychobiology 60 (2), 176-186

We look forward to receiving your revised manuscript.

Kind regards,

Elsa Addessi

Academic Editor

PLOS ONE

Journal Requirements:

Reviewers' comments:

Reviewer's Responses to Questions

**Comments to the Author**

1. If the authors have adequately addressed your comments raised in a previous round of review and you feel that this manuscript is now acceptable for publication, you may indicate that here to bypass the “Comments to the Author” section, enter your conflict of interest statement in the “Confidential to Editor” section, and submit your "Accept" recommendation.

Reviewer #1: (No Response)

2. Is the manuscript technically sound, and do the data support the conclusions?

Reviewer #1: Yes

3. Has the statistical analysis been performed appropriately and rigorously? 

Reviewer #1: Yes

4. Have the authors made all data underlying the findings in their manuscript fully available?

Reviewer #1: Yes

5. Is the manuscript presented in an intelligible fashion and written in standard English?

Reviewer #1: Yes

6. Review Comments to the Author

Reviewer #1: I would like to thank the authors for taking the Reviewers’ suggestions on board. I think the manuscript is in good shape now, and I only have a couple of minor comments.

No model stability assessment is provided, yet. The authors responded to my comment to provide a model stability assessment by saying they now selected the model with complete random structure. But this wasn’t the point of the question. My point related to information on influence diagnostics. The concern is that complex models such as these (including 3-way and several 2-way interaction terms) on data that stem from a small number of cases might result in low model stability, meaning the results could be unreliable even if the model converged and tests to compare the effects of the predictor terms reveal significant effects. If influence diagnostics should reveal that this was the case, it’s still interesting to report the results, but they should be accompanied by a note of caution that model stability was limited.

Results

Exp. 1: More appropriate to start with reporting the interaction, because otherwise the reader is tempted to think these main effects are interpretable on their own. Lines 401-404: its misleading to say that you found main effects of value & probability of risky option because these variables explain the results only in relation to other variables in the interactions. These lines can be deleted.

Exp.2: Move sentence 449-451 up to beginning of the paragraph (l 443), to fill the reader in what the final model looks like, before reading about the model results in more detail.

Comparison Exp1 and Exp2: like for Exp. 1, please don’t report main effects that are part of significant interactions as stand-alone significant effects; they are not interpretable outside the interaction (l 491-494). In this paragraph (l 491-497), you switch between talking about final and full model; is it necessary here to refer to the non-significant interaction in the full model (l 496)?

Latency analysis: l 510-511: “Finally, we analysed the response times in both experiments (to assess whether the higher levels of risk-proneness in Experiment 2 could be due to high impulsivity levels)” -> Could you elaborate on this a bit, it’s not clear why you suspect potentially more impulsivity in Exp. 2 than in Exp. 1.

It didn’t catch my eye in previous rounds, but apparently no transformation was applied to the response time for the analysis. Usually, raw data of response times don’t meet model assumptions and its common to log-transform them for analysis. Was this not necessary here, were raw response times sufficiently normally distributed?

L 446: replace investigation with investigating

L452: increased -> increase

In general, I would suggest to slightly re-phrase the result descriptions throughout, to make the readers life easier. Instead of referring to the significance of an effect, describe the effect and report the stats behind the statement.

For example, instead of saying “the main effects of the probability and the value of the risky option were significant (respectively χ2(1)=13.40, p<.001 and χ2(1)=47.66, p<.001)”, re-phrase to something along the lines of “subjects picked the risky cup more/less often with increasing probability of the risky option (χ2(1)=13.40, p<.001) and more/less often with increasing value of the risky option (χ2(1)=47.66, p<.001).

7. PLOS authors have the option to publish the peer review history of their article (what does this mean?). If published, this will include your full peer review and any attached files.

Reviewer #1: No

---

## [Editor Report · Decision Letter 4]

6 Oct 2022

PONE-D-21-29220R4Rationality and cognitive bias in captive gorillas' and orang-utans' economic decision-makingPLOS ONE

Dear Dr. Lacombe,

Thank you for submitting your manuscript to PLOS ONE. After careful consideration, we feel that it has merit but does not fully meet PLOS ONE’s publication criteria as it currently stands. Therefore, we invite you to submit a revised version of the manuscript that addresses the points raised during the review process.

Please carefully address the following minimal changes required before the manuscript can be accepted: 

L 402, 444, 501 “Cook's distances”

L 403 please delete a space before the semicolon

L 443 please replace “as there were” with “as they were”

L 444, 501  “expect” should be “except”

L 445, 502 please delete a space after Ketawa and before the semicolon; Cook's should be with initial capital letter

L 463 “increase” rather than “increased”

LL 523-5 “as the large, risky reward was shown to the subjects before each trial in this design only which could have led subjects to choose that risky reward”: this sentence looks problematic, please clarify

L 631 “induced”

L 633 “for an example”

L 665 please move “[62]” after children

L 666 “an exploitation”

L 667 “an exploration”

L 698 please replace the comma with a semicolon

Please proof-read the entire article once more, as I kept spotting some typos.

We look forward to receiving your revised manuscript.

Kind regards,

Elsa Addessi

Academic Editor

PLOS ONE
---

## [Editor Report · Decision Letter 5]

11 Nov 2022

Rationality and cognitive bias in captive gorillas' and orang-utans' economic decision-making

PONE-D-21-29220R5

Dear Dr. Lacombe,

We’re pleased to inform you that your manuscript has been judged scientifically suitable for publication and will be formally accepted for publication once it meets all outstanding technical requirements.

Kind regards,

Elsa Addessi

Academic Editor

PLOS ONE
---

## [Editor Report · Acceptance letter]

23 Nov 2022

PONE-D-21-29220R5 

Rationality and cognitive bias in captive gorillas' and orang-utans' economic decision-making 

Dear Dr. Lacombe:

I'm pleased to inform you that your manuscript has been deemed suitable for publication in PLOS ONE. Congratulations! Your manuscript is now with our production department. 

Kind regards, 

on behalf of

Dr. Elsa Addessi 

Academic Editor

PLOS ONE